# Painless Stochastic Gradient:
# Interpolation, Line-Search, and Convergence Rates

**Sharan Vaswani**
Mila, Université de Montréal

**Aaron Mishkin**
University of British Columbia

**Issam Laradji**
University of British Columbia
Element AI

**Mark Schmidt**
University of British Columbia, 1QBit
CCAI Affiliate Chair (Amii)

**Gauthier Gidel**
Mila, Université de Montréal
Element AI

**Simon Lacoste-Julien**[†]
Mila, Université de Montréal

## Abstract

Recent works have shown that stochastic gradient descent (SGD) achieves the fast convergence rates of full-batch gradient descent for over-parameterized models satisfying certain interpolation conditions. However, the step-size used in these works depends on unknown quantities and SGD's practical performance heavily relies on the choice of this step-size. We propose to use line-search techniques to automatically set the step-size when training models that can interpolate the data. In the interpolation setting, we prove that SGD with a stochastic variant of the classic Armijo line-search attains the deterministic convergence rates for both convex and strongly-convex functions. Under additional assumptions, SGD with Armijo line-search is shown to achieve fast convergence for non-convex functions. Furthermore, we show that stochastic extra-gradient with a Lipschitz line-search attains linear convergence for an important class of non-convex functions and saddle-point problems satisfying interpolation. To improve the proposed methods' practical performance, we give heuristics to use larger step-sizes and acceleration. We compare the proposed algorithms against numerous optimization methods on standard classification tasks using both kernel methods and deep networks. The proposed methods result in competitive performance across all models and datasets, while being robust to the precise choices of hyper-parameters. For multi-class classification using deep networks, SGD with Armijo line-search results in both faster convergence and better generalization.

## 1   Introduction

Stochastic gradient descent (SGD) and its variants [18, 21, 35, 39, 72, 82, 87] are the preferred optimization methods in modern machine learning. They only require the gradient for one training example (or a small "mini-batch" of examples) in each iteration and thus can be used with large datasets. These first-order methods have been particularly successful for training highly-expressive, over-parameterized models such as non-parametric regression [7, 45] and deep neural networks [9, 88]. However, the practical efficiency of stochastic gradient methods is adversely affected by two challenges: (i) their performance heavily relies on the choice of the step-size ("learning rate") [9, 70] and (ii) their slow convergence compared to methods that compute the full gradient (over all training examples) in each iteration [58].

[†] Canada CIFAR AI Chair..

Variance-reduction (VR) methods [18, 35, 72] are relatively new variants of SGD that improve its slow convergence rate. These methods exploit the finite-sum structure of typical loss functions arising in machine learning, achieving both the low iteration cost of SGD and the fast convergence rate of deterministic methods that compute the full-gradient in each iteration. Moreover, VR makes setting the learning rate easier and there has been work exploring the use of line-search techniques for automatically setting the step-size for these methods [71, 72, 76, 81]. These methods have resulted in impressive performance on a variety of problems. However, the improved performance comes at the cost of additional memory [72] or computational [18, 35] overheads, making these methods less appealing when training high-dimensional models on large datasets. Moreover, in practice VR methods do not tend to converge faster than SGD on over-parameterized models [19].

Indeed, recent works [5, 13, 33, 47, 52, 73, 83] have shown that when training over-parameterized models, classic SGD with a constant step-size and *without VR* can achieve the convergence rates of full-batch gradient descent. These works assume that the model is expressive enough to *interpolate* the data. The interpolation condition is satisfied for models such as non-parametric regression [7, 45], over-parametrized deep neural networks [88], boosting [69], and for linear classifiers on separable data. However, the good performance of SGD in this setting relies on using the proposed constant step-size, which depends on problem-specific quantities not known in practice. On the other hand, there has been a long line of research on techniques to automatically set the step-size for classic SGD. These techniques include using meta-learning procedures to modify the main stochastic algorithm [2, 6, 63, 75, 77, 86, 86], heuristics to adjust the learning rate on the fly [20, 43, 70, 74], and recent adaptive methods inspired by online learning [21, 39, 51, 60, 67, 68, 87]. However, none of these techniques have been proved to achieve the fast convergence rates that we now know are possible in the over-parametrized setting.

In this work, we use classical line-search methods [59] to automatically set the step-size for SGD when training over-parametrized models. Line-search is a standard technique to adaptively set the step-size for deterministic methods that evaluate the full gradient in each iteration. These methods make use of additional function/gradient evaluations to characterize the function around the current iterate and adjust the magnitude of the descent step. The additional noise in SGD complicates the use of line-searches in the general stochastic setting and there have only been a few attempts to address this. Mahsereci et al. [53] define a Gaussian process model over probabilistic Wolfe conditions and use it to derive a termination criterion for the line-search. The convergence rate of this procedure is not known, and experimentally we found that our proposed line-search technique is simpler to implement and more robust. Other authors [12, 17, 22, 42, 62] use a line-search termination criteria that requires function/gradient evaluations averaged over multiple samples. However, in order to achieve convergence, the number of samples required per iteration (the "batch-size") increases progressively, losing the low per iteration cost of SGD. Other work [11, 26] exploring trust-region methods assume that the model is sufficiently accurate, which is not guaranteed in the general stochastic setting. In contrast to these works, our line-search procedure does not consider the general stochastic setting and is designed for models that satisfy interpolation; it achieves fast rates in the over-parameterized regime without the need to manually choose a step-size or increase the batch size.

We make the following contributions: in Section 3 we prove that, under interpolation, SGD with a stochastic variant of the Armijo line-search attains the convergence rates of full-batch gradient descent in both the convex and strongly-convex settings. We achieve these rates under weaker assumptions than the prior work [83] and *without* the explicit knowledge of problem specific constants. We then consider minimizing non-convex functions satisfying interpolation [5, 83]. Previous work [5] proves that constant step-size SGD achieves a linear rate for non-convex functions satisfying the PL inequality [37, 65]. SGD is further known to achieve deterministic rates for general non-convex functions under a stronger assumption on the growth of the stochastic gradients [73, 83]. Under this assumption and an upper bound (that requires knowledge of the "Lipschitz" constant) on the maximum step size, we prove that SGD with Armijo line-search can achieve the deterministic rate for general non-convex functions (Section 4). Note that these are the first convergence rates for SGD with line-search in the interpolation setting for both convex and non-convex functions.

Moving beyond SGD, in Section 5 we consider the stochastic extra-gradient (SEG) method [24, 31, 36, 41, 55] used to solve general variational inequalities [27]. These problems encompass both convex minimization and saddle point problems arising in robust supervised learning [8, 84] and learning with non-separable losses or regularizers [4, 34]. In the interpolation setting, we show that a variant of SEG [24] with a "Lipschitz" line-search convergences linearly when minimizing an

important class of non-convex functions [16, 40, 44, 78, 79] satisfying the restricted secant inequality (RSI). Moreover, in Appendix E, we prove that the same algorithm results in linear convergence for both strongly convex-concave and bilinear saddle point problems satisfying interpolation.

In Section 6, we give heuristics to use large step-sizes and integrate acceleration with our line-search techniques, which improves practical performance of the proposed methods. We compare our algorithms against numerous optimizers [21, 39, 51, 53, 60, 68] on a synthetic matrix factorization problem (Section 7.2), convex binary-classification problems using radial basis function (RBF) kernels (Section 7.3), and non-convex multi-class classification problems with deep neural networks (Section 7.4). We observe that when interpolation is (approximately) satisfied, the proposed methods are robust and have competitive performance across models and datasets. Moreover, SGD with Armijo line-search results in both faster convergence and better generalization performance for classification using deep networks. Finally, in Appendix G.2, we evaluate SEG with line-search for synthetic bilinear saddle point problems. The code to reproduce our results can be found at https://github.com/IssamLaradji/sls.

We note that in concurrent work to ours, Berrada et al. [10] propose adaptive step-sizes for SGD on convex, finite-sum loss functions under an $\epsilon$-interpolation condition. Unlike our approach, $\epsilon$-interpolation requires knowledge of a lower bound on the global minimum and only guarantees approximate convergence to a stationary point. Moreover, in order to obtain linear convergence rates, they assume $\mu$-strong-convexity of *each* individual function. This assumption with $\epsilon$-interpolation reduces the finite-sum optimization to minimization of *any single* function in the finite sum.

## 2 Assumptions

We aim to minimize a differentiable function $f$ assuming access to noisy stochastic gradients of the function. We focus on the common machine learning setting where the function $f$ has a *finite-sum structure* meaning that $f(w) = \frac{1}{n} \sum_{i=1}^{n} f_i(w)$. Here $n$ is equal to the number of points in the training set and the function $f_i$ is the loss function for the training point $i$. Depending on the model, $f$ can either be strongly-convex, convex, or non-convex. We assume that $f$ is lower-bounded by some value $f^*$ and that $f$ is $L$-smooth [56] implying that the gradient $\nabla f$ is $L$-Lipschitz continuous.

We assume that the model is able to interpolate the data and use this property to derive convergence rates. Formally, interpolation requires that the gradient with respect to *each* point converges to zero at the optimum, implying that if the function $f$ is minimized at $w^*$ and thus $\nabla f(w^*) = 0$, then for all functions $f_i$ we have that $\nabla f_i(w^*) = 0$. For example, interpolation is exactly satisfied when using a linear model with the squared hinge loss for binary classification on linearly separable data.

## 3 Stochastic Gradient Descent for Convex Functions

Stochastic gradient descent (SGD) computes the gradient of the loss function corresponding to one or a mini-batch of randomly (typically uniformly) chosen training examples $i_k$ in iteration $k$. It then performs a descent step as $w_{k+1} = w_k - \eta_k \nabla f_{ik}(w_k)$, where $w_{k+1}$ and $w_k$ are the SGD iterates, $\eta_k$ is the step-size and $\nabla f_{ik}(\cdot)$ is the (average) gradient of the loss function(s) chosen at iteration $k$. Each stochastic gradient $\nabla f_{ik}(w)$ is assumed to be unbiased, implying that $\mathbb{E}_i [\nabla f_i(w)] = \nabla f(w)$ for all $w$. We now describe the Armijo line-search method to set the step-size in each iteration.

### 3.1 Armijo line-search

Armijo line-search [3] is a standard method for setting the step-size for gradient descent in the deterministic setting [59]. We adapt it to the stochastic case as follows: at iteration $k$, the Armijo line-search selects a step-size satisfying the following condition:

$$f_{ik} (w_k - \eta_k \nabla f_{ik}(w_k)) \leq f_{ik}(w_k) - c \cdot \eta_k \|\nabla f_{ik}(w_k)\|^2 . \tag{1}$$

Here, $c > 0$ is a hyper-parameter. Note that the above line-search condition uses the function and gradient values *of the mini-batch* at the current iterate $w_k$. Thus, compared to SGD, checking this condition only makes use of additional mini-batch function (and not gradient) evaluations. In the context of deep neural networks, this corresponds to extra forward passes on the mini-batch.

In our theoretical results, we assume that there is a maximum step-size $\eta_{\max}$ from which the line-search starts in *each* iteration $k$ and that we choose the largest step-size $\eta_k$ (less than or equal to $\eta_{\max}$) satisfying (1). In practice, backtracking line-search is a common way to ensure that Equation 1 is satisfied. Starting from $\eta_{\max}$, backtracking iteratively decreases the step-size by a constant factor

$\beta$ until the line-search succeeds (see Algorithm 1). Suitable strategies for *resetting* the step-size can avoid backtracking in the majority of iterations and make the step-size selection procedure efficient. We describe such strategies in Section 6. With resetting, we required (on average) only one additional forward pass on the mini-batch per iteration when training a standard deep network model (Section 7.4). Empirically, we observe that the algorithm is robust to the choice of both $c$ and $\eta_{\max}$; setting $c$ to a small constant and $\eta_{\max}$ to a large value consistently results in good performance.

We bound the chosen step-size in terms of the properties of the function(s) selected in iteration $k$.

**Lemma 1.** *The step-size $\eta_k$ returned by the Armijo line-search and constrained to lie in the $(0, \eta_{max}]$ range satisfies the following inequality,*

$$\eta_k \geq \min\left\{ \frac{2\,(1-c)}{L_{ik}}, \eta_{max} \right\}, \tag{2}$$

*where $L_{ik}$ is the Lipschitz constant of $\nabla f_{i_k}$.*

The proof is in Appendix A and follows the deterministic case [59]. Note that Equation (1) holds for all smooth functions (for small-enough $\eta_k$), does not require convexity, and guarantees backtracking line-search will terminate at a non-zero step-size. The parameter $c$ controls the "aggressiveness" of the algorithm; small $c$ values encourage a larger step-size. For a sufficiently large $\eta_{\max}$ and $c \leq 1/2$, the step-size is at least as large as $1/L_{ik}$, which is the constant step-size used in the interpolation setting [73, 83]. In practice, we expect these larger step-sizes to result in improved performance. In Appendix A, we also give upper bounds on $\eta_k$ if the function $f_{i_k}$ satisfies the Polyak-Lojasiewicz (PL) inequality [37, 65] with constant $\mu_{ik}$. PL is a weaker condition than strong-convexity and does not require convexity. In this case, $\eta_k$ is upper-bounded by the minimum of $\eta_{\max}$ and $1/(2c \cdot \mu_{ik})$. If we use a backtracking line-search that multiplies the step-size by $\beta$ until (1) holds, the step-size will be smaller by at most a factor of $\beta$ (we do not include this dependence in our results).

### 3.2 Convergence rates

In this section, we characterize the convergence rate of SGD with Armijo line-search in the strongly-convex and convex cases. The theorems below are proved in Appendix B and Appendix C respectively.

**Theorem 1** (Strongly-Convex). *Assuming (a) interpolation, (b) $L_i$-smoothness, (c) convexity of $f_i$'s, and (d) $\mu$ strong-convexity of $f$, SGD with Armijo line-search with $c = 1/2$ in Eq. 1 achieves the rate:*

$$\mathbb{E}\left[ \|w_T - w^*\|^2 \right] \leq \max\left\{ \left(1 - \frac{\bar{\mu}}{L_{max}}\right), (1 - \bar{\mu}\,\eta_{max}) \right\}^T \|w_0 - w^*\|^2.$$

*Here $\bar{\mu} = \sum_{i=1}^{n} \mu_i / n$ is the average strong-convexity of the finite sum and $L_{max} = \max_i L_i$ is the maximum smoothness constant in the $f_i$'s.*

In contrast to the previous results [52, 73, 83] that depend on $\mu$, the above linear rate depends on $\bar{\mu} \leq \mu$. Note that unlike Berrada et al. [10], we do not require that *each* $f_i$ is strongly convex, but for $\bar{\mu}$ to be non-zero we still require that *at least one* of the $f_i$'s is strongly-convex.

**Theorem 2** (Convex). *Assuming (a) interpolation, (b) $L_i$-smoothness and (c) convexity of $f_i$'s, SGD with Armijo line-search for all $c > 1/2$ in Equation 1 and iterate averaging achieves the rate:*

$$\mathbb{E}\left[ f(\bar{w}_T) - f(w^*) \right] \leq \frac{c \cdot \max\left\{ \frac{L_{max}}{2\,(1-c)}, \frac{1}{\eta_{max}} \right\}}{(2c - 1)\,T} \|w_0 - w^*\|^2.$$

*Here, $\bar{w}_T = \frac{\left[\sum_{i=1}^{T} w_i\right]}{T}$ is the averaged iterate after $T$ iterations and $L_{max} = \max_i L_i$.*

In particular, setting $c = 2/3$ implies that $\mathbb{E}\left[ f(\bar{w}_T) - f(w^*) \right] \leq \frac{\max\left\{ 3\,L_{\max}, \frac{2}{\eta_{\max}} \right\}}{T} \|w_0 - w^*\|^2$. These are the first rates for SGD with line-search in the interpolation setting and match the corresponding rates for full-batch gradient descent on strongly-convex and convex functions. This shows SGD attains fast convergence under interpolation *without* explicit knowledge of the Lipschitz constant. Next, we use the above line-search to derive convergence rates of SGD for non-convex functions.

## 4 Stochastic Gradient Descent for Non-convex Functions

To prove convergence results in the non-convex case, we additionally require the strong growth condition (SGC) [73, 83] to hold. The function $f$ satisfies the SGC with constant $\rho$, if $\mathbb{E}_i \|\nabla f_i(w)\|^2 \leq$

$\rho \|\nabla f(w)\|^2$ holds for any point $w$. This implies that if $\nabla f(w) = 0$, then $\nabla f_i(w) = 0$ for *all* $i$. Thus, functions satisfying the SGC necessarily satisfy the interpolation property. The SGC holds for all smooth functions satisfying a PL condition [83]. Under the SGC, we show that by upper-bounding the maximum step-size $\eta_{max}$, SGD with Armijo line-search achieves an $O(1/T)$ convergence rate.

**Theorem 3** (Non-convex). *Assuming (a) the SGC with constant $\rho$ and (b) $L_i$-smoothness of $f_i$'s, SGD with Armijo line-search in Equation 1 with $c = 1/2$ and setting $\eta_{max} = {}^3/_{2\rho L}$ achieves the rate:*

$$\min_{k=0,...,T-1} \mathbb{E} \|\nabla f(w_k)\|^2 \le \frac{4\, L_{max}}{T} \left( \frac{2\rho}{3} + 1 \right) \left( f(w_0) - f(w^*) \right).$$

We prove Theorem 3 in Appendix D. The result requires knowledge of $\rho\, L_{\max}$ to bound the maximum step-size, which is less practically appealing. It is not immediately clear how to relax this condition and we leave it for future work. However, in the next section, we show that if the non-convex function satisfies a specific curvature condition, a modified stochastic extra-gradient algorithm can achieve a linear rate under interpolation without additional assumptions or knowledge of the Lipschitz constant.

# 5 Stochastic Extra-Gradient Method

In this section, we use a modified stochastic extra-gradient (SEG) method for convex and non-convex minimization. For finite-sum minimization, stochastic extra-gradient (SEG) has the following update:

$$w'_k = w_k - \eta_k \nabla f_{ik}(w_k)\,,\; w_{k+1} = w_k - \eta_k \nabla f_{ik}(w'_k). \tag{3}$$

It computes the gradient at an extrapolated point $w'_k$ and uses it in the update from the current iterate $w_k$. Note that using the same sample $i_k$ and step-size $\eta_k$ for both steps [24] is important for the subsequent theoretical results. We now describe a "Lipschitz" line-search strategy [31, 32, 38] in order to automatically set the step-size for SEG.

## 5.1 Lipschitz line-search

The "Lipschitz" line-search has been used by previous work in the deterministic [32, 38] and the variance reduced settings [30]. It selects a step-size $\eta_k$ that satisfies the following condition:

$$\|\nabla f_{ik}(w_k - \eta_k \nabla f_{ik}(w_k)) - \nabla f_{ik}(w_k)\| \le c\, \|\nabla f_{ik}(w_k)\|. \tag{4}$$

As before, we use backtracking line-search starting from the maximum value of $\eta_{\max}$ to ensure that the chosen step-size satisfies the above condition. If the function $f_{ik}$ is $L_{ik}$-smooth, the step-size returned by the Lipschitz line-search satisfies $\eta_k \ge \min \{{}^c/_{L_{ik}}, \eta_{\max}\}$. Like the Armijo line-search in Section 3, the Lipschitz line-search does not require knowledge of the Lipschitz constant. Unlike the line-search strategy in the previous sections, checking condition (4) requires computing the gradient at a prospective extrapolation point. We now prove convergence rates for SEG with Lipschitz line-search for both convex and a special class of non-convex problems.

## 5.2 Convergence rates for minimization

For the next result, we assume that each function $f_i(\cdot)$ satisfies the restricted secant inequality (RSI) with constant $\mu_i$, implying that for all $w$, $\langle \nabla f_i(w), w - w^* \rangle \ge \mu_i \|w - w^*\|^2$. RSI is a weaker condition than strong-convexity. With additional assumptions, RSI is satisfied by important non-convex models such as single hidden-layer neural networks [40, 44, 78], matrix completion [79] and phase retrieval [16]. Under interpolation, we show SEG results in linear convergence for functions satisfying RSI. In particular, we obtain the following guarantee:

**Theorem 4** (Non-convex + RSI). *Assuming (a) interpolation, (b) $L_i$-smoothness, and (c) $\mu_i$-RSI of $f_i$'s, SEG with Lipschitz line-search in Eq. 4 with $c = {}^1/_4$ and $\eta_{max} \le \min_i 1/4\mu_i$ achieves the rate:*

$$\mathbb{E}\left[\|w_T - \mathcal{P}_{\mathcal{X}^*}[w_T]\|^2\right] \le \max \left\{ \left(1 - \frac{\bar{\mu}}{4\,L_{max}}\right), (1 - \eta_{max}\,\bar{\mu}) \right\}^T \|w_0 - \mathcal{P}_{\mathcal{X}^*}[w_0]\|^2,$$

*where $\bar{\mu} = \frac{\sum_{i=1}^n \mu_i}{n}$ is the average RSI constant of the finite sum and $\mathcal{X}^*$ is the non-empty set of optimal solutions. The operation $\mathcal{P}_{\mathcal{X}^*}[w]$ denotes the projection of $w$ onto $\mathcal{X}^*$.*

See Appendix E.2 for proof. Similar to the result of Theorem 1, the rate depends on the average RSI constant. Note that we do not require explicit knowledge of the Lipschitz constant to achieve the above rate. The constraint on the maximum step-size is mild since the minimum $\mu_i$ is typically small, thus allowing for large step-sizes. Moreover, Theorem 4 improves upon the $\left(1 - \mu^2/L^2\right)$ rate

| **Algorithm 1** SGD+Armijo$(f, w_0, \eta_{\max}, b, c, \beta, \gamma, \mathtt{opt})$ | **Algorithm 2** reset$(\eta, \eta_{\max}, \gamma, b, k, \mathtt{opt})$ |
|---|---|
| 1: **for** $k = 0, \ldots, T$ **do** | 1: **if** k = 1 **then** |
| 2: $\quad i_k \leftarrow$ sample mini-batch of size $b$ | 2: $\quad$ **return** $\eta_{\max}$ |
| 3: $\quad \eta \leftarrow \mathtt{reset}(\eta, \eta_{\max}, \gamma, b, k, \mathtt{opt})/\beta$ | 3: **else if** $\mathtt{opt} = 0$ **then** |
| 4: $\quad$ **repeat** | 4: $\quad \eta \leftarrow \eta$ |
| 5: $\quad\quad \eta \leftarrow \beta \cdot \eta$ | 5: **else if** $\mathtt{opt} = 1$ **then** |
| 6: $\quad\quad \tilde{w}_k \leftarrow w_k - \eta \nabla f_{ik}(w_k)$ | 6: $\quad \eta \leftarrow \eta_{\max}$ |
| 7: $\quad$ **until** $f_{ik}(\tilde{w}_k) \leq f_{ik}(w_k) - c \cdot \eta \, \|\nabla f_{ik}(w_k)\|^2$ | 7: **else if** $\mathtt{opt} = 2$ **then** |
| 8: $\quad w_{k+1} \leftarrow \tilde{w}_k$ | 8: $\quad \eta \leftarrow \eta \cdot \gamma^{b/n}$ |
| 9: **end for** | 9: **end if** |
| 10: **return** $w_{k+1}$ | 10: **return** $\eta$ |

Figure 1: Algorithm 1 gives pseudo-code for SGD with Armijo line-search. Algorithm 2 implements several heuristics (by setting $\mathtt{opt}$) for resetting the step-size at each iteration.

obtained using constant step-size SGD [5, 83]. In Appendix E.2, we show that the same rate can be attained by SEG with a constant step-size. In Appendix E.3, we show that under interpolation, SEG with Lipschitz line-search also achieves the desired $O(1/T)$ rate for convex functions.

### 5.3 Convergence rates for saddle point problems

In Appendix E.4, we use SEG with Lipschitz line-search for a class of saddle point problems of the form $\min_{u \in U} \max_{v \in \mathcal{V}} \phi(u, v)$. Here $\mathcal{U}$ and $\mathcal{V}$ are the constraint sets for the variables $u$ and $v$ respectively. In Theorem 6 in Appendix E.4, we show that under interpolation, SEG with Lipschitz line-search results in linear convergence for functions $\phi(u, v)$ that are strongly-convex in $u$ and strongly-concave in $v$. The required conditions are satisfied for robust optimization [84] with expressive models capable of interpolating the data. Furthermore, the interpolation property can be used to improve the convergence for a bilinear saddle-point problem [24, 25, 54, 85]. In Theorem 7 in Appendix E.5, we show that SEG with Lipschitz line-search results in linear convergence under interpolation. We empirically validate this claim with simple synthetic experiments in Appendix G.2.

## 6 Practical Considerations

In this section, we give heuristics to use larger step-sizes across iterations and discuss ways to use common acceleration schemes with our line-search techniques.

### 6.1 Using larger step-sizes

Recall that our theoretical analysis assumes that the line-search in *each* iteration starts from a global maximum step-size $\eta_{\max}$. However, in practice, this strategy increases the amount of backtracking and consequently the algorithm's runtime. A simple alternative is to initialize the line-search in each iteration to the step-size selected in the previous iteration $\eta_{k-1}$. With this strategy, the step-size can not increase and convergence is slowed in practice (it takes smaller steps than necessary). To alleviate these problems, we consider increasing the step-size across iterations by initializing the backtracking at iteration $k$ with $\eta_{k-1} \cdot \gamma^{b/n}$ [71, 72], where $b$ is the size of the mini-batch and $\gamma > 1$ is a tunable parameter. These heuristics correspond to the options used in Algorithm 2.

We also consider the Goldstein line-search that uses additional function evaluations to check the curvature condition $f_{ik}(w_k - \eta_k \nabla f_{ik}(w_k)) \geq f_{ik}(w_k) - (1 - c) \cdot \eta_k \|\nabla f_{ik}(w_k)\|^2$ and increases the step-size if it is not satisfied. Here, $c$ is the constant in Equation 1. The resulting method decreases the step-size if the Armijo condition is not satisfied and increases it if the curvature condition does not hold. Algorithm 3 in Appendix H gives pseudo-code for SGD with the Goldstein line-search.

### 6.2 Acceleration

In practice, augmenting stochastic methods with some form of momentum or acceleration [57, 64] often results in faster convergence [80]. Related work in this context includes algorithms specifically designed to achieve an accelerated rate of convergence in the stochastic setting [1, 23, 46]. Unlike these works, we propose simple ways of using either Polyak [64] or Nesterov [57] acceleration with the proposed line-search techniques. In both cases, similar to adaptive methods using momentum [80], we use SGD with Armijo line-search to determine $\eta_k$ and then use it directly within the acceleration scheme. When using Polyak momentum, the effective update can be given as: $w_{k+1} = w_k - $

$\eta_k \nabla f_{ik}(w_k) + \alpha(w_k - w_{k-1})$, where $\alpha$ is the momentum factor. This update rule has been used with a constant step-size and proven to obtain linear convergence rates on the *generalization error* for quadratic functions under an interpolation condition [48, 49]. For Nesterov acceleration, we use the variant for the convex case [57] (which has no additional hyper-parameters) with our line-search. The pseudo-code for using these methods with the Armijo line-search is given in Appendix H.

# 7 Experiments

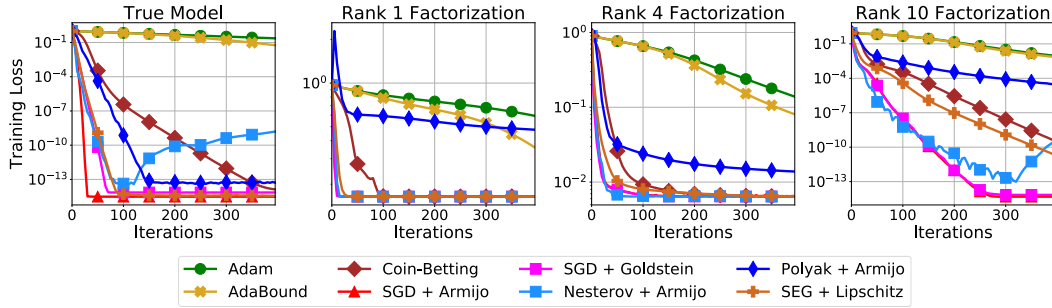

Figure 2: Matrix factorization using the true model and rank 1, 4, 10 factorizations. Rank 1 factorization is under-parametrized, while ranks 4 and 10 are over-parametrized. Rank 10 and the true model satisfy interpolation.

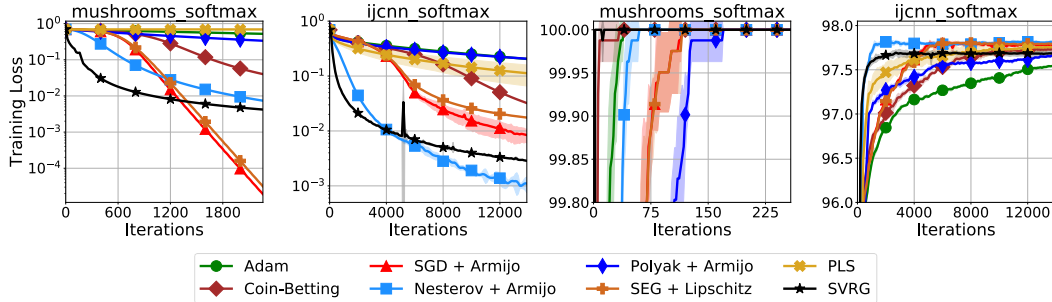

Figure 3: Binary classification using a softmax loss and RBF kernels for the mushrooms and ijcnn datasets. Mushrooms is linear separable in kernel-space with the selected kernel bandwidths while ijcnn is *not*. Overall, we observe fast convergence of SGD + Armijo, Nesterov + Armijo, and SEG + Lipschitz for both datasets.

We describe the experimental setup in Section 7.1. In Section 7.2, we present synthetic experiments to show the benefits of over-parametrization. In Sections 7.3 and 7.4, we showcase the convergence and generalization performance of our methods for kernel experiments and deep networks, respectively.

## 7.1 Experimental setup

We benchmark five configurations of the proposed line-search methods: SGD with (1) Armijo line-search with resetting the initial step-size (Algorithm 1 using option 2 in Algorithm 2), (2) Goldstein line-search (Algorithm 3), (3) Polyak momentum (Algorithm 5), (4) Nesterov acceleration (Algorithm 6), and (5) SEG with Lipschitz line-search (Algorithm 4) with option 2 to reset the step-size. Appendix F gives additional details on our experimental setup and the default hyper-parameters used for the proposed line-search methods. We compare our methods against Adam [39], which is the most common adaptive method, and other methods that report better performance than Adam: coin-betting [60], L4[1] [68], and Adabound [51]. We use the default learning rates for the competing methods. Unless stated otherwise, our results are averaged across 5 independent runs.

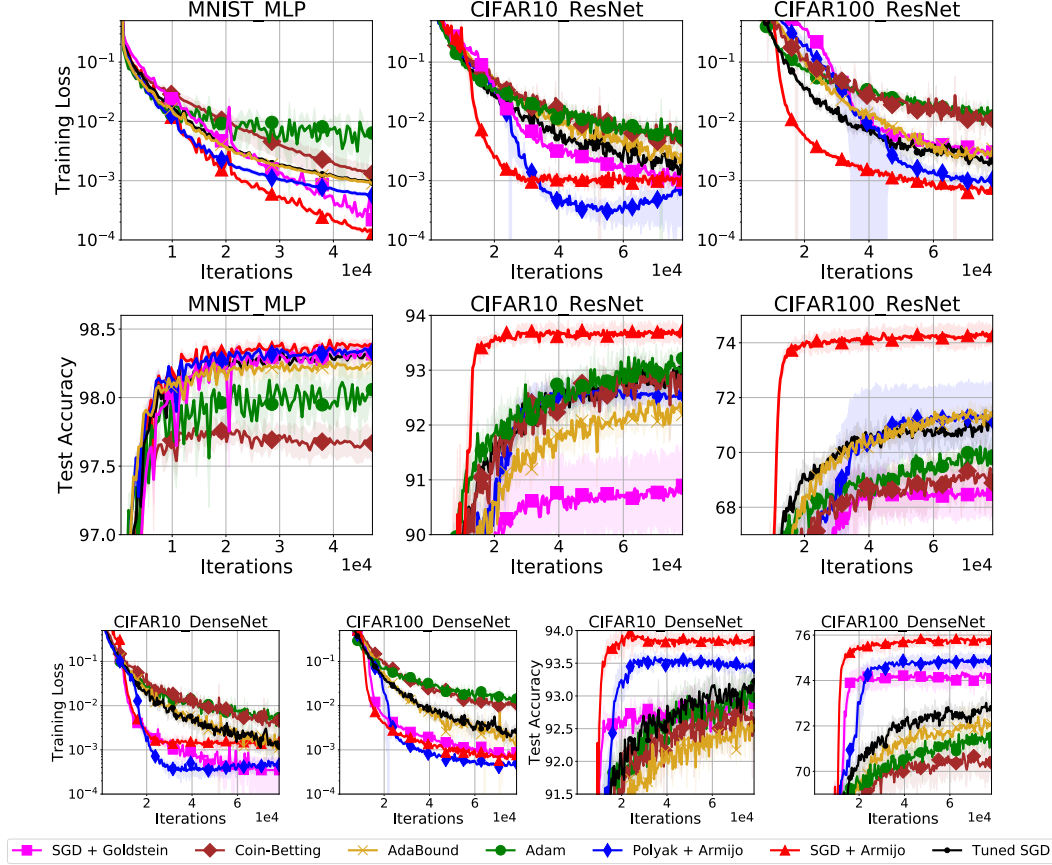

Figure 4: Multi-class classification using softmax loss and (top) an MLP model for MNIST; ResNet model for CIFAR-10 and CIFAR-100 (bottom) DenseNet model for CIFAR-10 and CIFAR-100.

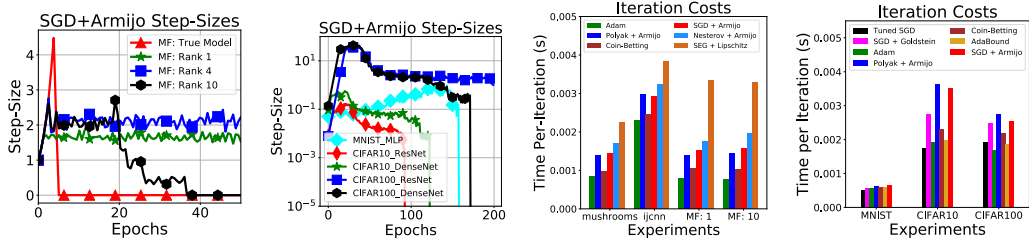

Figure 5: (Left) Variation in step-sizes for SGD+Armijo for the matrix factorization problem and classification with deep neural networks. (Right) Average time per iteration.

## 7.2 Synthetic experiment

We examine the effect of over-parametrization on convergence rates for the non-convex regression problem: $\min_{W_1, W_2} \mathbb{E}_{x \sim N(0,I)} \|W_2 W_1 x - A x\|^2$. This is equivalent to a matrix factorization problem satisfying RSI [79] and has been proposed as a challenging benchmark for gradient descent methods [66]. Following Rolínek et al. [68], we choose $A \in \mathbb{R}^{10 \times 6}$ with condition number $\kappa(A) = 10^{10}$ and generate a fixed dataset of 1000 samples. Unlike the previous work, we consider stochastic optimization and control the model's expressivity via the rank $k$ of the matrix factors $W_1 \in \mathbb{R}^{k \times 6}$ and $W_2 \in \mathbb{R}^{10 \times k}$. Figure 2 shows plots of training loss (averaged across 20 runs) for the true data-generating model, and using factors with rank $k \in \{1, 4, 10\}$.

We make the following observations: (i) for $k = 4$ (where interpolation *does not hold*) the proposed methods converge quicker than other optimizers but all methods reach an artificial optimization floor, (ii) using $k = 10$ yields an over-parametrized model where SGD with both Armijo and Goldstein

line-search converge linearly to machine precision, (iii) SEG with Lipschitz line-search obtains fast convergence according to Theorem 4, and (iv) adaptive-gradient methods stagnate in all cases. These observations validate our theoretical results and show that over-parameterization and line-search can allow for fast, "painless" optimization using SGD and SEG.

## 7.3 Binary classification with kernels

We consider convex binary classification using RBF kernels without regularization. We experiment with four standard datasets: mushrooms, rcv1, ijcnn, and w8a from LIBSVM [14]. The mushrooms dataset satisfies the interpolation condition with the selected kernel bandwidths, while ijcnn, rcv1, and w8a do not. For these experiments we also compare against a standard VR method (SVRG) [35] and probabilistic line-search (PLS) [53].[2] Figure 3 shows the training loss and test accuracy on mushrooms and ijcnn for the different optimizers with softmax loss. Results for rcv1 and w8a are given in Appendix G.3. We make the following observations: (i) SGD + Armijo, Nesterov + Armijo, and SEG + Lipschitz perform the best and are comparable to hand-tuned SVRG. (ii) The proposed line-search methods perform well on ijcnn even though it is not separable in kernel space. This demonstrates some robustness to violations of the interpolation condition.

## 7.4 Multi-class classification using deep networks

We benchmark the convergence rate and generalization performance of our line-search methods on standard deep learning experiments. We consider non-convex minimization for multi-class classification using deep network models on the MNIST, CIFAR10, and CIFAR100 datasets. Our experimental choices follow the setup in Luo et al. [51]. For MNIST, we use a 1 hidden-layer multi-layer perceptron (MLP) of width 1000. For CIFAR10 and CIFAR100, we experiment with the standard image-classification architectures: ResNet-34 [28] and DenseNet-121 [29]. We also compare to the best performing constant step-size SGD with the step-size selected by grid search.

From Figure 4, we observe that: (i) SGD with Armijo line-search consistently leads to the best performance in terms of both the training loss and test accuracy. It also converges to a good solution *much* faster when compared to the other methods. (ii) The performance of SGD with line-search and Polyak momentum is always better than "tuned" constant step-size SGD and Adam, whereas that of SGD with Goldstein line-search is competitive across datasets. We omit Nesterov + Armijo as it unstable and diverges and omit SEG since it resulted in slower convergence and worse performance.

We also verify that our line-search methods do not lead to excessive backtracking and function evaluations. Figure 5 (right) shows the cost per iteration for the above experiments. Our line-searches methods are only marginally slower than Adam and converge much faster. In practice, we observed SGD+Armijo uses only one additional function evaluation on average. Figure 5 (left) shows the evolution of step-sizes for SGD+Armijo in our experiments. For deep neural networks, SGD+Armijo automatically finds a step-size schedule resembling cosine-annealing [50]. In Appendix G.1, we evaluate and compare the hyper-parameter sensitivity of Adam, constant step-size SGD, and SGD with Armijo line-search on CIFAR10 with ResNet-34. While SGD is sensitive to the choice of the step-size, the performance of SGD with Armijo line-search is robust to the value of $c$ in the $[0.1, 0.5]$ range. There is virtually no effect of $\eta_{\max}$, since the correct range of step-sizes is found in early iterations.

## 8 Conclusion

We showed that under the interpolation condition satisfied by modern over-parametrized models, simple line-search techniques for classic SGD and SEG lead to fast convergence in both theory and practice. For future work, we hope to strengthen our results for non-convex minimization using SGD with line-search and study stochastic momentum techniques under interpolation. More generally, we hope to utilize the rich literature on line-search and trust-region methods to improve stochastic optimization for machine learning.

## Acknowledgments

We would like to thank Yifan Sun and Nicolas Le Roux for insightful discussions. AM is supported by the NSERC CGS M award. IL is funded by the UBC Four-Year Doctoral Fellowships (4YF), This research was also partially supported by the Canada CIFAR AI Chair Program, the CIFAR LMB Program, by a Google Focused Research award, by an IVADO postdoctoral scholarship (for SV), by a Borealis AI fellowship (for GG), by the Canada Excellence Research Chair in "Data Science for Realtime Decision-making" and by the NSERC Discovery Grants RGPIN-2017-06936 and 2015-06068.

## Footnotes

[1]L4 applied to momentum SGD (L4 Mom) in `https://github.com/iovdin/l4-pytorch` was unstable in our experiments and we omit it from the main paper.

[2]PLS is impractical for deep networks since it requires the second moment of the mini-batch gradients and needs GP model inference for every line-search evaluation.

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
