[Supplementary Material]

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

[3]Note that these choices are inspired by the theory

[4]We hope to use method such as [89] to automatically set the momentum parameter in the future.

[5]This code is available at `https://github.com/benjamin-recht/shallow-linear-net`

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

# A   Proof of Lemma 1

*Proof.*

From the smoothness of $f_{ik}$ and the update rule, the following inequality holds for all values of $\eta_k$.

$$f_{ik}(w_{k+1}) \leq f_{ik}(w_k) - \left(\eta_k - \frac{L_{ik}\eta_k^2}{2}\right) \|\nabla f_{ik}(w_k)\|^2$$

The step-size returned by the line-search satisfies Equation 1, implying that,

$$f_{ik}(w_{k+1}) \leq f_{ik}(w_k) - c\,\eta_k \|\nabla f_{ik}(w_k)\|^2$$

Using the above relations, the step-size returned by the line-search satisfies the following inequality,

$$c\,\eta_k \geq \left(\eta_k - \frac{L_{ik}\eta_k^2}{2}\right)$$

$$\implies \eta_k \geq \frac{2\,(1-c)}{L_{ik}}$$

This gives us a lower bound on $\eta_k$.

Let us now upper-bound $\eta_k$. Using Equation 1,

$$\implies \eta_k \leq \frac{[f_{ik}(w_k) - f_{ik}(w_{k+1})]}{c\,\|\nabla f_{ik}(x_k)\|^2}$$

$$\eta_k \leq \frac{[f_{ik}(w_k) - f_{ik}(w^*) + f_{ik}(w^*) - f_{ik}(w_{k+1})]}{c\,\|\nabla f_{ik}(w_k)\|^2}$$

By the interpolation condition, $f_{ik}(w^*) \leq f_{ik}(w)$ for all functions $i_k$ and points $w$, $\implies f_{ik}(w^*) - f_{ik}(w_{k+1}) \leq 0$

$$\implies \eta_k \leq \frac{[f_{ik}(w_k) - f_{ik}(w^*)]}{c\,\|\nabla f_{ik}(w_k)\|^2}$$

By definition, $\eta_k \leq \eta_{\max}$. Furthermore, if we each $f_{ik}(\cdot)$ satisfies the either strong-convexity or the Polyak-Lojasiewicz (PL) inequality [37, 65] (which is weaker than strong-convexity and does not require convexity), then,

$$f_{ik}(w_k) - f_{ik}(w_k^*) \leq \frac{1}{2\mu_{ik}} \|\nabla f_{ik}(w_k)\|^2$$

$$\implies f_{ik}(w_k) - f_{ik}(w_{k+1}) \leq \frac{1}{2\mu_{ik}} \|\nabla f_{ik}(w_k)\|^2$$

$$f_{ik}(w_k) - f_{ik}(w^*) \leq \frac{1}{2\mu_{ik}} \|\nabla f_{ik}(w_k)\|^2 \qquad \text{(Using the interpolation condition)}$$

$$\implies f_{ik}(w_k) - f_{ik}(w_{k+1}) \leq \frac{1}{2\mu_{ik}} \|\nabla f_{ik}(w_k)\|^2 \qquad \text{(Since, } f_{ik}(w^*) \leq f_{ik}(w_{k+1}).\text{)}$$

$$\implies c \cdot \eta_k \leq \frac{\|\nabla f_{ik}(w_k)\|^2}{2\mu_{ik} \|\nabla f_{ik}(w_k)\|^2} \qquad \text{(From the above relation on } \eta_k.\text{)}$$

$$\implies c \cdot \eta_k \leq \frac{1}{2\mu_{ik}}$$

Thus, the step-size returned by the line-search satisfies the relation $\eta_k \leq \min\{\frac{1}{2c\cdot\mu_{ik}}, \eta_{\max}\}$.

From the above relations,

$$\eta_k \in \left[\min\left\{\frac{2\,(1-c)}{L_{ik}}, \eta_{\max}\right\}, \min\left\{\frac{1}{2c\cdot\mu_{ik}}, \eta_{\max}\right\}\right]$$

# B  Proof for Theorem 1

*Proof.*

$$\|w_{k+1} - w^*\|^2 = \|w_k - \eta_k \nabla f_{ik}(w_k) - w^*\|^2$$
$$\|w_{k+1} - w^*\|^2 = \|w_k - w^*\|^2 - 2\eta_k \langle \nabla f_{ik}(w_k), w_k - w^* \rangle + \eta_k^2 \|\nabla f_{ik}(w_k)\|^2$$

Using strong-convexity of $f_{ik}(\cdot)$ (and setting $\mu_{ik} = 0$ if the $f_{i_k}$ is not strongly-convex),

$$-\langle \nabla f_{ik}(w_k), w_k - w^* \rangle \leq f_{ik}(w^*) - f_{ik}(w_k) - \frac{\mu_{ik}}{2} \|w_k - w^*\|^2$$

$$\implies \|w_{k+1} - w^*\|^2 \leq \|w_k - w^*\|^2 + 2\eta_k \left[ f_{ik}(w^*) - f_{ik}(w_k) - \frac{\mu_{ik}}{2} \|w_k - w^*\|^2 \right] + \eta_k^2 \|\nabla f_{ik}(w_k)\|^2$$

$$= \|w_k - w^*\|^2 + 2\eta_k \left[ f_{ik}(w^*) - f_{ik}(w_k) \right] - \mu_{ik}\eta_k \|w_k - w^*\|^2 + \eta_k^2 \|\nabla f_{ik}(w_k)\|^2$$

$$\implies \|w_{k+1} - w^*\|^2 \leq (1 - \mu_{ik}\eta_k) \|w_k - w^*\|^2 + 2\eta_k \left[ f_{ik}(w^*) - f_{ik}(w_k) \right] + \eta_k^2 \|\nabla f_{ik}(w_k)\|^2$$

Using Equation 1,

$$\eta_k^2 \|\nabla f_{ik}(w_k)\|^2 \leq \frac{\eta_k}{c} \left[ f_{ik}(w_k) - f_{ik}(w_{k+1}) \right]$$

$$\implies \|w_{k+1} - w^*\|^2 \leq (1 - \mu_{ik}\eta_k) \|w_k - w^*\|^2 + 2\eta_k \left[ f_{ik}(w^*) - f_{ik}(w_k) \right] + \frac{\eta_k}{c} \left[ f_{ik}(w_k) - f_{ik}(w_{k+1}) \right]$$

The interpolation condition implies that $w^*$ is the minimum for all functions $f_i$, implying that for all $i$, $f_i(w^*) \leq f_i(w_{k+1})$.

$$\|w_{k+1} - w^*\|^2 \leq (1 - \mu_{ik}\eta_k) \|w_k - w^*\|^2 + 2\eta_k \left[ f_{ik}(w^*) - f_{ik}(w_k) \right] + \frac{\eta_k}{c} \left[ f_{ik}(w_k) - f_{ik}(w^*) \right]$$

$$= (1 - \mu_{ik}\eta_k) \|w_k - w^*\|^2 + \left( 2\eta_k - \frac{\eta_k}{c} \right) \left[ f_{ik}(w^*) - f_{ik}(w_k) \right]$$

The term $[f_{ik}(w^*) - f_{ik}(w_k)]$ is negative. Let $c \geq \frac{1}{2} \implies \left( 2\eta_k - \frac{\eta_k}{c} \right) \geq 0$ for all $\eta_k$.

$$\implies \|w_{k+1} - w^*\|^2 \leq (1 - \mu_{ik}\eta_k) \|w_k - w^*\|^2$$

Taking expectation wrt to $i_k$,

$$\implies \mathbb{E} \left[ \|w_{k+1} - w^*\|^2 \right] \leq \mathbb{E}_{ik} \left[ (1 - \mu_{ik}\eta_k) \|w_k - w^*\|^2 \right]$$

$$= (1 - \mathbb{E}_{ik} [\mu_{ik}\eta_k]) \|w_k - w^*\|^2$$

$$\leq \left( 1 - \mathbb{E}_{ik} \left[ \mu_{ik} \ \min \left\{ \frac{2(1-c)}{L_{ik}}, \eta_{\max} \right\} \right] \right) \|w_k - w^*\|^2 \qquad \text{(Using Equation 2)}$$

Setting $c = 1/2$,

$$\implies \mathbb{E} \left[ \|w_{k+1} - w^*\|^2 \right] \leq \left( 1 - \mathbb{E}_{ik} \left[ \mu_{ik} \ \min \left\{ \frac{1}{L_{ik}}, \eta_{\max} \right\} \right] \right) \|w_k - w^*\|^2$$

We consider the following two cases: $\eta_{\max} < 1/L_{\max}$ and $\eta_{\max} \geq 1/L_{\max}$. When $\eta_{\max} < 1/L_{\max}$, we have $\eta_{\max} < 1/L_{ik}$ and,

$$\mathbb{E} \left[ \|w_{k+1} - w^*\|^2 \right] \leq (1 - \mathbb{E}_{ik} [\mu_{ik} \ \eta_{\max}]) \|w_k - w^*\|^2$$

$$= (1 - \mathbb{E}_{ik} [\mu_{ik}] \ \eta_{\max}) \|w_k - w^*\|^2 = (1 - \bar{\mu} \ \eta_{\max}) \|w_k - w^*\|^2$$

By recursion through iterations $k = 1$ to $T$,

$$\mathbb{E} \left[ \|w_T - w^*\|^2 \right] \leq (1 - \bar{\mu} \ \eta_{\max})^T \|w_0 - w^*\|^2 .$$

When $\eta_{\max} \geq 1/L_{\max}$, we use $\min\left\{\frac{1}{L_{ik}}, \eta_{\max}\right\} \geq \min\left\{\frac{1}{L_{\max}}, \eta_{\max}\right\}$ to obtain

$$\mathbb{E}\left[\|w_{k+1} - w^*\|^2\right] \leq \left(1 - \mathbb{E}_{ik}\left[\mu_{ik} \ \min\left\{\frac{1}{L_{\max}}, \eta_{\max}\right\}\right]\right)\|w_k - w^*\|^2$$

$$= \left(1 - \mathbb{E}_{ik}\left[\mu_{ik} \ \frac{1}{L_{\max}}\right]\right)\|w_k - w^*\|^2$$

$$= \left(1 - \frac{\mathbb{E}_{ik}[\mu_{ik}]}{L_{\max}}\right)\|w_k - w^*\|^2 = \left(1 - \frac{\bar{\mu}}{L_{\max}}\right)\|w_k - w^*\|^2$$

By recursion through iterations $k = 1$ to $T$,

$$\mathbb{E}\left[\|w_T - w^*\|^2\right] \leq \left(1 - \frac{\bar{\mu}}{L_{\max}}\right)^T \|w_0 - w^*\|^2.$$

Putting the two cases together,

$$\mathbb{E}\left[\|w_T - w^*\|^2\right] \leq \max\left\{\left(1 - \frac{\bar{\mu}}{L_{\max}}\right), (1 - \bar{\mu} \ \eta_{\max})\right\}^T \|w_0 - w^*\|^2$$

$\square$

## C   Proof for Theorem 2

*Proof.*

$$\|w_{k+1} - w^*\|^2 = \|w_k - \eta_k \nabla f_{ik}(w_k) - w^*\|^2$$

$$\|w_{k+1} - w^*\|^2 = \|w_k - w^*\|^2 - 2\eta_k\langle\nabla f_{ik}(w_k), w_k - w^*\rangle + \eta_k^2\|\nabla f_{ik}(w_k)\|^2$$

$$2\eta_k\langle\nabla f_{ik}(w_k), w_k - w^*\rangle = \|w_k - w^*\|^2 - \|w_{k+1} - w^*\|^2 + \eta_k^2\|\nabla f_{ik}(w_k)\|^2$$

$$\langle\nabla f_{ik}(w_k), w_k - w^*\rangle = \frac{1}{2\eta_k}\left[\|w_k - w^*\|^2 - \|w_{k+1} - w^*\|^2\right] + \frac{\eta_k}{2}\|\nabla f_{ik}(w_k)\|^2$$

$$\leq \frac{1}{2\eta_k}\left[\|w_k - w^*\|^2 - \|w_{k+1} - w^*\|^2\right] + \frac{f_{ik}(w_k) - f_{ik}(w_{k+1})}{2c} \qquad \text{(Using Equation 1)}$$

The interpolation condition implies that $w^*$ is the minimum for all functions $f_i$, implying that for all $i$, $f_i(w^*) \leq f_i(w_{k+1})$.

$$\implies \langle\nabla f_{ik}(w_k), w_k - w^*\rangle \leq \frac{1}{2\eta_k}\left[\|w_k - w^*\|^2 - \|w_{k+1} - w^*\|^2\right] + \frac{f_{ik}(w_k) - f_{ik}(w^*)}{2c}$$

Taking expectation wrt $i_k$,

$$\mathbb{E}\left[\langle\nabla f_{ik}(w_k), w_k - w^*\rangle\right] \leq \mathbb{E}\left[\frac{1}{2\eta_k}\left[\|w_k - w^*\|^2 - \|w_{k+1} - w^*\|^2\right]\right] + \mathbb{E}\left[\frac{f_{ik}(w_k) - f_{ik}(w^*)}{2c}\right]$$

$$= \mathbb{E}\left[\frac{1}{2\eta_k}\left[\|w_k - w^*\|^2 - \|w_{k+1} - w^*\|^2\right]\right] + \left[\frac{f(w_k) - f(w^*)}{2c}\right]$$

$$\implies \langle\mathbb{E}\left[\nabla f_{ik}(w_k)\right], w_k - w^*\rangle \leq \mathbb{E}\left[\frac{1}{2\eta_k}\left[\|w_k - w^*\|^2 - \|w_{k+1} - w^*\|^2\right]\right] + \left[\frac{f(w_k) - f(w^*)}{2c}\right]$$

$$\implies \langle\nabla f(w_k), w_k - w^*\rangle \leq \mathbb{E}\left[\frac{1}{2\eta_k}\left[\|w_k - w^*\|^2 - \|w_{k+1} - w^*\|^2\right]\right] + \left[\frac{f(w_k) - f(w^*)}{2c}\right]$$

By convexity,

$$f(w_k) - f(w^*) \leq \langle\nabla f(w_k), w_k - w^*\rangle$$

$$\implies f(w_k) - f(w^*) \leq \mathbb{E}\left[\frac{1}{2\eta_k}\left[\|w_k - w^*\|^2 - \|w_{k+1} - w^*\|^2\right]\right] + \left[\frac{f(w_k) - f(w^*)}{2c}\right]$$

If $1 - \frac{1}{2c} \geq 0 \implies$ if $c \geq \frac{1}{2}$, then

$$\implies f(w_k) - f(w^*) \leq \mathbb{E}\left[\frac{c}{(2c-1)\eta_k}\left[\|w_k - w^*\|^2 - \|w_{k+1} - w^*\|^2\right]\right]$$

Taking expectation and summing from $k = 0$ to $k = T - 1$

$$\implies \mathbb{E}\left[\sum_{k=0}^{T-1}[f(w_k) - f(w^*)]\right] \leq \mathbb{E}\left[\sum_{k=0}^{T-1}\frac{c}{(2c-1)\eta_k}\left[\|w_k - w^*\|^2 - \|w_{k+1} - w^*\|^2\right]\right]$$

By Jensen's inequality,

$$\mathbb{E}\left[f(\bar{w}_T) - f(w^*)\right] \leq \mathbb{E}\left[\sum_{k=0}^{T-1}\left[\frac{f(w_k) - f(w^*)}{T}\right]\right]$$

$$\implies \mathbb{E}\left[f(\bar{w}_T) - f(w^*)\right] \leq \frac{1}{T}\mathbb{E}\left[\sum_{k=0}^{T-1}\frac{c}{(2c-1)\eta_k}\left[\|w_k - w^*\|^2 - \|w_{k+1} - w^*\|^2\right]\right]$$

If $\Delta_k = \|w_k - w^*\|^2$, then

$$\mathbb{E}\left[f(\bar{w}_T) - f(w^*)\right] \leq \frac{c}{T(2c-1)}\mathbb{E}\left[\sum_{k=0}^{T-1}\frac{1}{\eta_k}[\Delta_k - \Delta_{k+1}]\right]$$

Using Equation 2,

$$\frac{1}{\eta_k} \leq \max\left\{\frac{L_{ik}}{2(1-c)}, \frac{1}{\eta_{\max}}\right\} \leq \max\left\{\frac{L_{\max}}{2(1-c)}, \frac{1}{\eta_{\max}}\right\}$$

$$\implies \mathbb{E}\left[f(\bar{w}_T) - f(w^*)\right] \leq \frac{c \cdot \max\left\{\frac{L_{\max}}{2(1-c)}, \frac{1}{\eta_{\max}}\right\}}{(2c-1)T}\mathbb{E}\sum_{k=0}^{T-1}[\Delta_k - \Delta_{k+1}]$$

$$= \frac{c \cdot \max\left\{\frac{L_{\max}}{2(1-c)}, \frac{1}{\eta_{\max}}\right\}}{(2c-1)T}\mathbb{E}[\Delta_0 - \Delta_T]$$

$$\mathbb{E}\left[f(\bar{w}_T) - f(w^*)\right] \leq \frac{c \cdot \max\left\{\frac{L_{\max}}{2(1-c)}, \frac{1}{\eta_{\max}}\right\}}{(2c-1)T}\|w_0 - w^*\|^2$$

$\square$

# D   Proof for Theorem 3

*Proof.*

By the smoothness assumption,

$$f(w_{k+1}) \leq f(w_k) - \langle\nabla f(w_k), \eta_k\nabla f_{ik}(w_k)\rangle + \frac{L\eta_k^2}{2}\|\nabla f_{ik}(w_k)\|^2$$

$$\frac{f(w_{k+1}) - f(w_k)}{\eta_k} \leq -\langle\nabla f(w_k), \nabla f_{ik}(w_k)\rangle + \frac{L\eta_k}{2}\|\nabla f_{ik}(w_k)\|^2$$

Taking expectation,

$$\mathbb{E}\left[\frac{f(w_{k+1}) - f(w_k)}{\eta_k}\right] \leq -\|\nabla f(w_k)\|^2 + \mathbb{E}\left[\frac{L\eta_k}{2}\|\nabla f_{ik}(w_k)\|^2\right]$$

$$\leq -\|\nabla f(w_k)\|^2 + \frac{L\eta_{\max}}{2}\mathbb{E}\left[\|\nabla f_{ik}(w_k)\|^2\right]$$

$$\implies \mathbb{E}\left[\frac{f(w_{k+1}) - f(w_k)}{\eta_k}\right] \leq -\|\nabla f(w_k)\|^2 + \frac{L\eta_{\max}\rho}{2}\|\nabla f(w_k)\|^2 \qquad \text{(By the SGC)}$$

$$\implies \left(1 - \frac{L\eta_{\max}\rho}{2}\right)\|\nabla f(w_k)\|^2 \leq \mathbb{E}\left[\frac{f(w_k) - f(w_{k+1})}{\eta_k}\right]$$

If $\eta_{\max} \leq \frac{2}{L\rho}$,

$$\|\nabla f(w_k)\|^2 \leq \frac{1}{1 - \frac{L\eta_{\max}\rho}{2}}\mathbb{E}\left[\frac{f(w_k) - f(w_{k+1})}{\eta_k}\right]$$

If $f(w_k) - f(w_{k+1}) \leq 0$,

$$\implies \|\nabla f(w_k)\|^2 \leq 0 \implies \|\nabla f(w_k)\|^2 = 0$$

If $f(w_k) - f(w_{k+1}) \geq 0$,

$$\implies \|\nabla f(w_k)\|^2 \leq \frac{1}{1 - \frac{L\eta_{\max}\rho}{2}} \mathbb{E}\left[\frac{1}{\eta_k}\left(f(w_k) - f(w_{k+1})\right)\right]$$

From the line-search we know that,

$$\eta_k \geq \min\left\{\eta_{\max}, \frac{2(1-c)}{L_{ik}}\right\}$$

$$\implies \frac{1}{\eta_k} \leq \max\left\{\frac{1}{\eta_{\max}}, \frac{L_{ik}}{2(1-c)}\right\}$$

$$\leq \frac{1}{\eta_{\max}} + \frac{L_{ik}}{2(1-c)}$$

$$\implies \frac{1}{\eta_k} \leq \frac{1}{\eta_{\max}} + \frac{L_{max}}{2(1-c)}$$

From the above relations,

$$\implies \|\nabla f(w_k)\|^2 \leq \left(\frac{1}{1 - \frac{L\eta_{\max}\rho}{2}}\right)\left(\frac{1}{\eta_{\max}} + \frac{L_{max}}{2(1-c)}\right)\mathbb{E}\left[f(w_k) - f(w_{k+1})\right]$$

$$\implies \min_{k\in[T]}\mathbb{E}\|\nabla f(w_k)\|^2 \leq \frac{1}{T}\left(\frac{1}{1 - \frac{L\eta_{\max}\rho}{2}}\right)\left(\frac{1}{\eta_{\max}} + \frac{L_{max}}{2(1-c)}\right)\mathbb{E}\left[f(w_0) - f(w_T)\right]$$

$$\leq \frac{1}{T}\left(\frac{1}{1 - \frac{L\eta_{\max}\rho}{2}}\right)\left(\frac{1}{\eta_{\max}} + \frac{L_{max}}{2(1-c)}\right)(f(w_0) - f(w^*))$$

Let $c = 1/2$ and $\eta_{\max} = \frac{\gamma}{\rho L}$, for $\gamma < 2$,

$$\min_{k\in[T]}\mathbb{E}\|\nabla f(w_k)\|^2 \leq \frac{1}{T}\left(\frac{1}{1 - \frac{\gamma}{2}}\right)\left(\frac{\rho L}{\gamma} + L_{max}\right)(f(w_0) - f(w^*)).$$

Noting $L \leq L_{max}$,

$$\min_{k\in[T]}\mathbb{E}\|\nabla f(w_k)\|^2 \leq \frac{2\,L_{max}}{T}\left(\frac{1}{2 - \gamma}\right)\left(\frac{\rho}{\gamma} + 1\right)(f(w_0) - f(w^*))$$

Setting $\gamma = 3/2$,

$$\min_{k \in [T]} \mathbb{E} \left\| \nabla f(w_k) \right\|^2 \leq \frac{4 \, L_{max}}{T} \left( \frac{2\rho}{3} + 1 \right) (f(w_0) - f(w^*))$$

$\square$

# E   Proofs for SEG

## E.1   Common lemmas

We denote $\|u - v\|^2$ as $\Delta(u, v) = \Delta(v, u)$. We first prove the following lemma that will be useful in the subsequent analysis.

**Lemma 2.** *For any set of vectors $a, b, c, d$, if $a = b + c$, then,*

$$\Delta(a, d) = \Delta(b, d) - \Delta(a, b) + 2\langle c, a - d \rangle$$

*Proof.*

$$\begin{aligned}
\Delta(a, d) = \|a - d\|^2 &= \|b + c - d\|^2 \\
&= \|b - d\|^2 + 2\langle c, b - d \rangle + \|c\|^2
\end{aligned}$$

Since $c = a - b$,

$$\begin{aligned}
\Delta(a, d) &= \|b - d\|^2 + 2\langle a - b, b - d \rangle + \|a - b\|^2 \\
&= \|b - d\|^2 + 2\langle a - b, b - a + a - d \rangle + \|a - b\|^2 \\
&= \|b - d\|^2 + 2\langle a - b, b - a \rangle + 2\langle a - b, a - d \rangle + \|a - b\|^2 \\
&= \|b - d\|^2 - 2\|a - b\|^2 + 2\langle a - b, a - d \rangle + \|a - b\|^2 \\
&= \|b - d\|^2 - \|a - b\|^2 + 2\langle c, a - d \rangle \\
\Delta(a, d) &= \Delta(b, d) - \Delta(a, b) + 2\langle c, a - d \rangle.
\end{aligned}$$

$\square$

## E.2   Proof for Theorem 4

We start from Lemma 2 with $a = w_{k+1} = w_k - \eta_k \nabla f_{ik}(w_k')$ and $d = w^*$:

$$\begin{aligned}
\Delta(w_{k+1}, w^*) &= \Delta(w_k, w^*) - \Delta(w_{k+1}, w_k) - 2\eta_k \left[ \langle \nabla f_{ik}(w_k'), w_{k+1} - w^* \rangle \right]. \\
&= \Delta(w_k, w^*) - \eta_k^2 \left\| \nabla f_{ik}(w_k') \right\|^2 - 2\eta_k \left[ \langle \nabla f_{ik}(w_k'), w_{k+1} - w^* \rangle \right].
\end{aligned}$$

Using $w_{k+1} = w_k' + \eta_k \nabla f_{ik}(w_k) - \eta_k \nabla f_{ik}(w_k')$ and completing the square,

$$\begin{aligned}
\Delta(w_{k+1}, w^*) &= \Delta(w_k, w^*) - \eta_k^2 \left\| \nabla f_{ik}(w_k') \right\|^2 - 2\eta_k \left[ \langle \nabla f_{ik}(w_k'), w_k' + \eta_k \nabla f_{ik}(w_k) - \eta_k \nabla f_{ik}(w_k') - w^* \rangle \right] \\
&= \Delta(w_k, w^*) + \eta_k^2 \left\| \nabla f_{ik}(w_k') \right\|^2 - 2\eta_k \left[ \langle \nabla f_{ik}(w_k'), w_k' + \eta_k \nabla f_{ik}(w_k) - w^* \rangle \right] \\
&= \Delta(w_k, w^*) + \eta_k^2 \left\| \nabla f_{ik}(w_k') - \nabla f_{ik}(w_k) \right\|^2 - \eta_k^2 \left\| \nabla f_{ik}(w_k) \right\|^2 - 2\eta_k \left[ \langle \nabla f_{ik}(w_k'), w_k' - w^* \rangle \right]
\end{aligned}$$

Noting $\Delta(w_k', w_k) = \eta_k^2 \left\| \nabla f_{ik}(w_k) \right\|^2$ gives

$$\Delta(w_{k+1}, w^*) = \Delta(w_k, w^*) - \Delta(w_k', w_k) + \eta_k^2 \left\| \nabla f_{ik}(w_k') - \nabla f_{ik}(w_k) \right\|^2 - 2\eta_k \left[ \langle \nabla f_{ik}(w_k'), w_k' - w^* \rangle \right]$$

$$\implies 2\eta_k \left[ \langle \nabla f_{ik}(w_k'), w_k' - w^* \rangle \right] = \Delta(w_k, w^*) - \Delta(w_k', w_k) + \eta_k^2 \left\| \nabla f_{ik}(w_k') - \nabla f_{ik}(w_k) \right\|^2 - \Delta(w_{k+1}, w^*). \tag{5}$$

By RSI, which states that for all $w$, $\langle \nabla f_i(w), w - w^* \rangle \geq \mu_i \|w^* - w\|^2$, we have

$$\langle \nabla f_{ik}(w_k'), w_k' - w^* \rangle \geq \mu_{ik} \Delta(w_k', w^*)$$

By Young's inequality,

$$\Delta(w_k, w^*) \leq 2\Delta(w_k, w_k') + 2\Delta(w_k', w^*)$$
$$\implies 2\Delta(w_k', w^*) \geq \Delta(w_k, w^*) - 2\Delta(w_k, w_k')$$
$$\implies \langle 2\eta_k \nabla f_{ik}(w_k'), w_k' - w^* \rangle \geq \mu_{ik}\eta_k \left[ \Delta(w_k, w^*) - 2\Delta(w_k, w_k') \right]$$

Rearranging Equation (5),

$$\Delta(w_{k+1}, w^*) = \Delta(w_k, w^*) - \Delta(w_k', w_k) + \eta_k^2 \left\| \nabla f_{ik}(w_k') - \nabla f_{ik}(w_k) \right\|^2 - 2\eta_k \left[ \langle \nabla f_{ik}(w_k'), w_k' - w^* \rangle \right]$$
$$\implies \Delta(w_{k+1}, w^*) \leq \Delta(w_k, w^*) - \Delta(w_k', w_k) + \eta_k^2 \left\| \nabla f_{ik}(w_k') - \nabla f_{ik}(w_k) \right\|^2 - \mu_{ik}\eta_k \left[ \Delta(w_k, w^*) - 2\Delta(w_k, w_k') \right]$$
$$\Delta(w_{k+1}, w^*) \leq (1 - \eta_k \mu_{ik}) \Delta(w_k, w^*) - \Delta(w_k', w_k) + \eta_k^2 \left\| \nabla f_{ik}(w_k') - \nabla f_{ik}(w_k) \right\|^2 + 2\mu_{ik}\eta_k \Delta(w_k, w_k')$$

Now we consider using a constant step-size as well as the Lipschitz line-search.

### E.2.1    Using a constant step-size

*Proof.*

Using smoothness of $f_{ik}(\cdot)$,

$$\Delta(w_{k+1}, w^*) \leq (1 - \eta_k \mu_{ik}) \Delta(w_k, w^*) - \Delta(w_k', w_k) + \eta_k^2 L_{ik}^2 \Delta(w_k', w_k) + 2\mu_{ik}\eta_k \Delta(w_k, w_k')$$
$$\implies \Delta(w_{k+1}, w^*) \leq (1 - \eta_k \mu_{ik}) \Delta(w_k, w^*) + \left( \eta_k^2 L_{ik}^2 - 1 + 2\mu_{ik}\eta_k \right) \Delta(w_k', w_k)$$

Taking expectation with respect to $i_k$,

$$\mathbb{E} \left[ \Delta(w_{k+1}, w^*) \right] \leq \mathbb{E} \left[ (1 - \eta_k \mu_{ik}) \Delta(w_k, w^*) \right] + \mathbb{E} \left[ \left( \eta_k^2 L_{ik}^2 - 1 + 2\mu_{ik}\eta_k \right) \Delta(w_k', w_k) \right]$$

Note that $w_k$ doesn't depend on $i_k$. Furthermore, neither does $w^*$ because of the interpolation property.

$$\implies \mathbb{E} \left[ \Delta(w_{k+1}, w^*) \right] \leq \mathbb{E} \left[ 1 - \eta_k \mu_{ik} \right] \Delta(w_k, w^*) + \mathbb{E} \left[ \left( \eta_k^2 L_{ik}^2 - 1 + 2\mu_{ik}\eta_k \right) \Delta(w_k', w_k) \right]$$

If $\eta_k \leq \frac{1}{4 \cdot L_{\max}}$, then $\left( \eta_k^2 L_{ik}^2 - 1 + 2\mu_{ik}\eta_k \right) \leq 0$ and

$$\implies \mathbb{E} \left[ \Delta(w_{k+1}, w^*) \right] \leq \mathbb{E} \left[ 1 - \frac{\mu_{ik}}{4L_{\max}} \right] \Delta(w_k, w^*)$$
$$\implies \mathbb{E} \left[ \Delta(w_{k+1}, w^*) \right] \leq \left( 1 - \frac{\bar{\mu}}{4L_{\max}} \right) \Delta(w_k, w^*)$$
$$\implies \mathbb{E} \left[ \Delta(w_k, w^*) \right] \leq \left( 1 - \frac{\bar{\mu}}{4L_{\max}} \right)^T \Delta(w_0, w^*)$$

$\square$

### E.2.2    Using the line-search

*Proof.*

Using Equation (4) to control the difference in gradients,

$$\Delta(w_{k+1}, w^*) \leq (1 - \eta_k \mu_{ik}) \Delta(w_k, w^*) - \Delta(w_k', w_k) + c^2 \Delta(w_k', w_k) + 2\mu_{ik}\eta_k \Delta(w_k, w_k')$$
$$\implies \Delta(w_{k+1}, w^*) \leq (1 - \eta_k \mu_{ik}) \Delta(w_k, w^*) + \left( c^2 + 2\mu_{ik}\eta_k - 1 \right) \Delta(w_k', w_k)$$

Taking expectation with respect to $i_k$,

$$\mathbb{E} \left[ \Delta(w_{k+1}, w^*) \right] \leq \mathbb{E} \left[ 1 - \eta_k \mu_{ik} \Delta(w_k, w^*) \right] + \mathbb{E} \left[ \left( c^2 - 1 + 2\eta_k \mu_i \right) \Delta(w_k', w_k) \right]$$

Note that $w_k$ doesn't depend on $i_k$. Furthermore, neither does $w^*$ because of the interpolation property.

$$\implies \mathbb{E} \left[ \Delta(w_{k+1}, w^*) \right] \leq \mathbb{E} \left[ 1 - \eta_k \mu_{ik} \right] \Delta(w_k, w^*) + \mathbb{E} \left[ \left( c^2 - 1 + 2\eta_k \mu_{ik} \right) \Delta(w_k', w_k) \right]$$

Using smoothness, the line-search in Equation 4 is satisfied if $\eta_k \leq \frac{c}{L_{ik}}$, implying that the step-size returned by the line-search always satisfies $\eta_k \geq \min\left\{\frac{c}{L_{ik}}, \eta_{\max}\right\}$.

$$\implies \mathbb{E}\left[\Delta(w_{k+1}, w^*)\right] \leq \mathbb{E}\left(1 - \mu_{ik} \min\left\{\frac{c}{L_{ik}}, \eta_{\max}\right\}\right) \Delta(w_k, w^*) + \mathbb{E}\left[\left(c^2 - 1 + 2\eta_k \mu_{ik}\right) \Delta(w_k', w_k)\right]$$

If we ensure that $\eta_k \leq \frac{c}{\mu_{ik}}$, then $c^2 - 1 + 2\eta_k \mu_{ik} \leq 0$. In other words, we need to ensure that $\eta_{\max} \leq \min_i \frac{c}{\mu_i}$. Choosing $c = 1/4$, we obtain the following:

$$\mathbb{E}\left[\Delta(w_{k+1}, w^*)\right] \leq \mathbb{E}\left(1 - \mu_{ik} \min\left\{\frac{1}{4\ L_{ik}}, \eta_{\max}\right\}\right) \Delta(w_k, w^*)$$

We consider the following cases: $\eta_{\max} < \frac{1}{4\ L_{\max}}$ and $\eta_{\max} \geq \frac{1}{4\ L_{\max}}$. When $\eta_{\max} < \frac{1}{4\ L_{\max}}$,

$$\mathbb{E}\left[\Delta(w_{k+1}, w^*)\right] \leq \mathbb{E}\left(1 - \mu_{ik}\ \eta_{\max}\right) \Delta(w_k, w^*)$$
$$= \left(1 - \bar{\mu}\ \eta_{\max}\right) \Delta(w_k, w^*)$$
$$\implies \mathbb{E}\left[\Delta(w_{k+1}, w^*)\right] \leq \left(1 - \bar{\mu}\ \eta_{\max}\right)^T \Delta(w_0, w^*)$$

When $\eta_{\max} \geq 1/(4\ L_{\max})$, we use $\min\left\{\frac{1}{4\ L_{ik}}, \eta_{\max}\right\} \geq \min\left\{\frac{1}{4\ L_{\max}}, \eta_{\max}\right\}$ to obtain

$$\mathbb{E}\left[\Delta(w_{k+1}, w^*)\right] \leq \mathbb{E}\left(1 - \mu_{ik} \min\left\{\frac{1}{4\ L_{\max}}, \eta_{\max}\right\}\right) \Delta(w_k, w^*)$$
$$= \mathbb{E}\left(1 - \mu_{ik} \frac{1}{4\ L_{\max}}\right) \Delta(w_k, w^*)$$
$$= \left(1 - \frac{\bar{\mu}}{4\ L_{\max}}\right) \Delta(w_k, w^*)$$
$$\implies \mathbb{E}\left[\Delta(w_{k+1}, w^*)\right] \leq \left(1 - \frac{\bar{\mu}}{4\ L_{\max}}\right)^T \Delta(w_0, w^*).$$

Putting the two cases together, we obtain

$$\mathbb{E}\left[\Delta(w_{k+1}, w^*)\right] \leq \max\left\{\left(1 - \frac{\bar{\mu}}{4\ L_{\max}}\right), \left(1 - \bar{\mu}\ \eta_{\max}\right)\right\}^T \Delta(w_0, w^*).$$

$\square$

### E.3 Proof of SEG for convex minimization

**Theorem 5.** *Assuming the interpolation property and under L-smoothness and convexity of $f$, SEG with Lipschitz line-search with $c = 1/\sqrt{2}$ in Equation 4 and iterate averaging achieves the following rate:*

$$\mathbb{E}\left[f(\bar{w}_T) - f(w^*)\right] \leq \frac{2\ \max\left\{\sqrt{2}\ L_{max}, \frac{1}{\eta_{max}}\right\}}{T} \|w_0 - w^*\|^2\ .$$

*Here, $\bar{w}_T = \frac{\left[\sum_{i=1}^{T} w_i\right]}{T}$ is the averaged iterate after $T$ iterations.*

*Proof.* Starting from Equation (5),

$$2\eta_k \left[\langle \nabla f_{ik}(w_k'), w_k' - w^* \rangle\right] = \Delta(w_k, w^*) - \Delta(w_k', w_k) + \eta_k^2 \|\nabla f_{ik}(w_k') - \nabla f_{ik}(w_k)\|^2 - \Delta(w_{k+1}, w^*)$$

and using the standard convexity inequality,

$$\langle \nabla f_{ik}(w_k'), w_k' - w^* \rangle \geq f_{i_k}(w_k') - f_{i_k}(w^*)$$

$$\geq \tfrac{1}{4}(f_{i_k}(w_k') - f_{i_k}(w^*))$$

$$\geq \tfrac{1}{4}(f_{i_k}(w_k) - \eta_k \|\nabla f_{ik}(w_k)\|^2 - f_{i_k}(w^*))$$

$$= \tfrac{1}{4}(f_{i_k}(w_k) - \frac{1}{\eta_k}\Delta(w_k, w_k') - f_{i_k}(w^*))$$

$$\implies 2\eta_k \left[\langle \nabla f_{ik}(w_k'), w_k' - w^* \rangle\right] \geq \frac{\eta_k}{2}\left[f_{i_k}(w_k) - f_{i_k}(w^*)\right] - \frac{1}{2}\Delta(w_k, w_k')$$

where we used the interpolation hypothesis to say that $w^*$ is a minimizer of $f_{i_k}$ and thus $f_{i_k}(w_k') \geq f_{i_k}(w^*)$. Combining this with (5) and (4) leads to,

$$\frac{\eta_k}{2}(f_{i_k}(w_k) - f_{i_k}(w^*)) \leq \Delta(w_k, w^*) - \Delta(w_{k+1}, w^*) - \tfrac{1}{2}\Delta(w_k', w_k) + \eta_k^2 \|\nabla f_{ik}(w_k') - \nabla f_{ik}(w_k)\|^2$$

$$\leq \Delta(w_k, w^*) - \Delta(w_{k+1}, w^*) - (\tfrac{1}{2} - c^2)\Delta(w_k', w_k)$$

$$\leq \Delta(w_k, w^*) - \Delta(w_{k+1}, w^*),$$

$$\implies f_{i_k}(w_k) - f_{i_k}(w^*) \leq \frac{2}{\eta_k}\left[\Delta(w_k, w^*) - \Delta(w_{k+1}, w^*)\right]$$

where for the last inequality we used Equation 4 and the fact that $c^2 \leq 1/2$. By definition of the Lipschitz line-search, $\eta_k \in [\min\{c/L_{\max}, \eta_{\max}\}, \eta_{\max}]$, implying

$$\frac{1}{\eta_k} \leq \max\left\{\frac{L_{\max}}{c}, \frac{1}{\eta_{\max}}\right\}$$

Setting $c = \frac{1}{\sqrt{2}}$,

$$\frac{1}{\eta_k} \leq \max\left\{\sqrt{2}L_{\max}, \frac{1}{\eta_{\max}}\right\}$$

$$f_{i_k}(w_k) - f_{i_k}(w^*) \leq 2 \max\left\{\sqrt{2}L_{\max}, \frac{1}{\eta_{\max}}\right\}(\Delta(w_k, w^*) - \Delta(w_{k+1}, w^*))$$

Taking expectation with respect to $i_k$,

$$f(w_k) - f(w^*) \leq 2 \max\left\{\sqrt{2}L_{\max}, \frac{1}{\eta_{\max}}\right\}(\Delta(w_k, w^*) - \mathbb{E}\Delta(w_{k+1}, w^*))$$

Finally, taking the expectation respect to $w_k$ and summing for $k = 1, \ldots, T$, we get,

$$\mathbb{E}\left[f(\bar{w}_k) - f(w^*)\right] \leq \frac{2 \max\left\{\sqrt{2}L_{\max}, \frac{1}{\eta_{\max}}\right\} \Delta(w_0, w^*)}{T}$$

$\square$

### E.4 SEG for general strongly monotone operators

Let $F(\cdot)$ be a Lipschitz (strongly)-monotone operator. $F$ satisfies the following inequalities for all $u, v$,

$$\|F(u) - F(v)\| \leq L \|u - v\| \qquad \text{(Lipschitz continuity)}$$

$$\langle F(u) - F(v), u - v \rangle \geq \mu \|u - v\|^2 \qquad \text{(Strong monotonicity)}$$

Here, $\mu$ is the strong-monotonicity constant and $L$ is the Lipschitz constant. Note that $\mu = 0$ for monotone operators. We seek the solution $w^*$ to the following optimization problem: $\sup_w \langle F(w^*), w^* - w \rangle \leq 0$.

Note that for strongly-convex minimization where $w^* = \arg\min f(w)$, $F$ is equal to the gradient operator and $\mu$ and $L$ are the strong-convexity and smoothness constants in the previous sections.

SEG [36] is a common method for optimizing stochastic variational inequalities and results in an $O(1/\sqrt{T})$ rate for monotone operators and an $O(1/T)$ rate for strongly-monotone operators [24]. For strongly-monotone operators, the convergence can be improved to obtain a linear rate by using variance-reduction methods [15, 61] exploiting the finite-sum structure in $F$. In this setting,

$F(w) = \frac{1}{n}\sum_{i=1}^{n} F_i(w)$. To the best of our knowledge, the interpolation condition has not been studied in the context of general strongly monotone operators. In this case, the interpolation condition implies that $F_i(w^*) = 0$ for all operators $F_i$ in the finite sum.

**Theorem 6** (Strongly-monotone). *Assuming (a) interpolation, (b) L-smoothness and (c) $\mu$-strong monotonocity of $F$, SEG using Lipschitz line-search with $c = 1/4$ in Equation 4 and setting $\eta_{max} \leq \min_i \frac{1}{4\mu_i}$ has the rate:*

$$\mathbb{E}\left[\|w_k - w^*\|^2\right] \leq \left(\max\left\{\left(1 - \frac{\bar{\mu}}{4\,L_{max}}\right), (1 - \eta_{max}\,\bar{\mu})\right\}\right)^T \|w_0 - w^*\|^2 \,.$$

*Proof.*

For each $F_{ik}(\cdot)$, we use the strong-monotonicity condition with constant $\mu_{ik}$,

$$\langle F_{ik}(u) - F_{ik}(v), u - v\rangle \geq \mu_{ik}\,\|u - v\|^2$$

Set $u = w$, $v = w^*$,

$$\implies \langle F_{ik}(w) - F_{ik}(w^*), w - w^*\rangle \geq \mu_{ik}\,\|w - w^*\|^2$$

By the interpolation condition,

$$F_{ik}(w^*) = 0$$
$$\implies \langle F_{ik}(w), w - w^*\rangle \geq \mu_{ik}\,\|w - w^*\|^2$$

This is equivalent to an RSI-like condition, but with the gradient operator $\nabla f_{ik}(\cdot)$ replaced with a general operator $F_{ik}(()\cdot)$.

From here on, the theorem follows the same proof as that for Theorem 4 above with the $F_{ik}()$ instead of $\nabla f_{ik}()$ and the strong-convexity constant being replaced with the constant for strong-monotonicity. □

Like in the RSI case, the above result can also be obtained using a constant step-size $\eta \leq \frac{1}{4\,L_{\max}}$.

### E.5 SEG for bilinear saddle point problems

Let us consider the bilinear saddle-point problem of the form $\min_x \max_y x^\top Ay - x^\top b - y^\top c$, where $A$ is the "coupling" matrix and where both $b$ and $c$ are vectors [15, 24]. In this case, the (monotone) operator $F(x, y) = [Ax - b, -A^\top y + c]$ and we assume the finite sum formulation as:

$$x^\top Ay - x^\top b - y^\top c = \frac{1}{n}\sum_{i=0}^{n} x^\top A_i y - x^\top b_i - y^\top c_i \tag{6}$$

We show that the interpolation condition enables SEG with Lipschitz line-search achieve a linear rate of convergence. In every iteration, the SEG algorithm samples rows $A_i$ (resp. columns $A_j$) of the matrix $A$ and the respective coefficient $b_i$ (resp. $c_j$). If $x_k$ and $y_k$ correspond to the iterates for the minimization and maximization problem respectively, then the update rules for SEG can be written as:

$$\begin{cases} x_{k+1} = x_k - \eta_k(A_{i_k}y_{k+1/2} - b_{i_k}) \\ y_{k+1} = y_k + \eta_k(A_{i_k}^\top x_{k+1/2} - c_{i_k}) \end{cases} \text{and} \begin{cases} x_{k+1/2} = x_k - \eta_k(A_{i_k}y_k - b_{i_k}) \\ y_{k+1/2} = y_k + \eta_k(A_{i_k}^\top x_k - c_{i_k}) \end{cases} \tag{7}$$

which can be more compactly written as,

$$x_{k+1} = x_k - \eta_k(A_{i_k}(y_k + \eta_k(A_{i_k}^\top x_k - c_{i_k})) - b_{i_k}) \tag{8}$$
$$y_{k+1} = y_k + \eta_k(A_{i_k}^\top(x_k - \eta_k(A_{i_k}y_k - b_{i_k})) - c_{i_k}) \,.$$

We now prove that SEG attains the following linear rate of convergence.

**Theorem 7** (Bilinear). *Assuming the (a) interpolation property and for the (b) bilinear saddle point problem, SEG with Lipschitz line-search with $c = 1/\sqrt{2}$ in Equation 4 achieves the following rate:*

$$\mathbb{E}\left[\|w_k - w^*\|^2\right] \leq \left(\max\left\{\left(1 - \frac{\sigma_{\min}(\mathbb{E}[A_{i_k}A_{i_k}^\top])}{4\max_i \sigma_{max}(A_i A_i^\top)}\right), \left(1 - \frac{\eta_{max}}{2}\sigma_{\min}(\mathbb{E}[A_{i_k}A_{i_k}^\top])\right)\right\}\right)^T (\|x_k\|^2 + \|y_k\|^2)$$

*Proof.* If $(x^*, y^*)$ is the solution to the above saddle point problem, then interpolation hypothesis implies that

$$A_{i_k} y^* = b_{i_k} \quad \text{and} \quad A_i^\top x^* = c_i$$

We note that the problem can be reduced to the case $b = c = 0$ by using the change of variable $\tilde{x}_k := x_k - x^*$ and $\tilde{y}_k := y_k - y^*$.

$$\tilde{x}_{k+1} = x_{k+1} - x^* = x_k - x^* - \eta_k(A_{i_k}(y_k - y^* + \eta_k A_{i_k}^\top(x_k - x^*))) = \tilde{x}_k - \eta_k A_{i_k}(\tilde{y}_k + \eta_k A_{i_k}^\top \tilde{x}_k)$$

$$\tilde{y}_{k+1} = y_{k+1} - y^* = y_k - y^* + \eta_k(A_{i_k}^\top(x_k - x^* - \eta_k A_{i_k}(y_k - y^*))) = \tilde{y}_k + \eta_k A_{i_k}^\top(\tilde{x}_k - \eta_k A_{i_k}\tilde{y}_k)$$

Thus, $(\tilde{x}_{k+1}, \tilde{y}_{k+1})$ correspond to the update rule Eq.(8) with $b = c = 0$. Note that the interpolation hypothesis is key for this problem reduction.

In the following, without loss of generality, we will assume that $b = c = 0$.

Using the update rule, we get,

$$\|x_{k+1}\|^2 + \|y_{k+1}\|^2 = \|x_k\|^2 + \|y_k\|^2 - \eta_k^2(x_k^\top A_{i_k} A_{i_k}^\top x_k + y_k^\top A_{i_k}^\top A_{i_k} y_k) + \eta_k^4(x_k^\top (A_{i_k} A_{i_k}^\top)^2 x_k + y_k^\top (A_{i_k}^\top A_{i_k})^2 y_k)$$

The line-search hypothesis can be simplified as,

$$\eta_k^2(x_k^\top (A_{i_k} A_{i_k}^\top)^2 x_k + y_k^\top (A_{i_k}^\top A_{i_k})^2 y_k) \le c^2(x_k^\top A_{i_k} A_{i_k}^\top x_k + y_k^\top A_{i_k}^\top A_{i_k} y_k) \tag{9}$$

leading to,

$$\|x_{k+1}\|^2 + \|y_{k+1}\|^2 \le \|x_k\|^2 + \|y_k\|^2 - \eta_k^2(1 - c^2)(x_k^\top A_{i_k} A_{i_k}^\top x_k + y_k^\top A_{i_k}^\top A_{i_k} y_k)$$

Noting that $L_{\max} = \left[\max_i \sigma_{\max}(A_i A_i^\top)\right]^{1/2}$, we obtain $\eta_k \ge \min\left\{ \left[2\max_i \sigma_{\max}(A_i A_i^\top)\right]^{-1/2}, \eta_{\max} \right\}$ from the Lipschitz line-search. Taking the expectation with respect to $i_k$ gives,

$$\mathbb{E}\left[\|x_{k+1}\|^2 + \|y_{k+1}\|^2\right] \le (1 - \eta_k^2 \sigma_{\min}(\mathbb{E}[A_{i_k} A_{i_k}^\top])(1 - c^2))(\|x_k\|^2 + \|y_k\|^2)$$

$$\le \max\left\{ \left(1 - \frac{\sigma_{\min}(\mathbb{E}[A_{i_k} A_{i_k}^\top])}{4\max_i \sigma_{\max}(A_i A_i^\top)}\right), \left(1 - \frac{\eta_{\max}}{2} \sigma_{\min}(\mathbb{E}[A_{i_k} A_{i_k}^\top])\right) \right\}(\|x_k\|^2 + \|y_k\|^2).$$

Applying this inequality recursively and taking expectations yields the final result. $\qquad\square$

Observe that the rate depends on the minimum and maximum singular values of the matrix formed using the mini-batch of examples selected in the SEG iterations. Note that these are the first results for bilinear min-max problems in the stochastic, interpolation setting.

# F    Additional Experimental Details

In this section we give details for all experiments in the main paper and the additional results given in Appendix G. In all experiments, we used the default learning rates provided in the implementation for the methods we compare against. For the proposed line-search methods and for *all* experiments in this paper, we set the initial step-size $\eta_{\max} = 1$ and use back-tracking line-search where we reduce the step-size by a factor of $0.9$ if the line-search is not satisfied. We used $c = 0.1$ for all our experiments with both Armijo and Goldstein line-search procedures, $c = 0.9$ for SEG with Lipschitz line-search, and $c = 0.5$ when using Nesterov acceleration [3]. For Polyak acceleration, we use $c = 0.1$ in our experiments with deep neural networks and $c = 0.5$ otherwise. For our non-convex experiments, we always constrain the step-size to be less than $10$ to prevent it from becoming unbounded. Note that we conduct a robustness study to quantify the influence of the $c$ and $\eta_{\max}$ parameter in Section G.1. For the heuristic in [71, 72], we set the step-size increase factor to $\gamma = 1.5$ for convex minimization and use $\gamma = 2$ for non-convex minimization. Similarly, when using Polyak momentum we set the momentum factor to the highest value that does not lead to divergence. It is set to $\beta = 0.8$ in the convex case and $\beta = 0.6$ in the non-convex case [4].

## F.1    Synthetic Matrix Factorization Experiment

In the following we give additional details for synthetic matrix factorization experiment in Section 7.2. As stated in the main text, we set $A \in \mathbb{R}^{10 \times 6}$ with condition number $\kappa(A) = 10^{10}$ and generated a fixed dataset of 1000 samples using the code released by Ben Recht [5]. We withheld 200 of these examples as a test set. All optimizers used mini-batches of 100 examples and were run for 50 epochs. We averaged over 20 runs with different random seeds to control for variance in the training loss, which approached machine precision for several optimizers.

## F.2    Binary Classification using Kernel Methods

We give additional details for the experiments on binary classification with RBF kernels in Section 7.3. For all datasets, we used only the training sets available in the LIBSVM [14] library and used an 80:20 split of it. The 80 percent split of the data was used

| Dataset | Dimension ($d$) | Training Set Size | Test Set Size | Kernel Bandwidth | SVRG Step-Size |
|---|---|---|---|---|---|
| mushrooms | 112 | 6499 | 1625 | 0.5 | 500 |
| ijcnn | 22 | 39992 | 9998 | 0.05 | 500 |
| rcv1 | 47236 | 16194 | 4048 | 0.25 | 500 |
| w8a | 300 | 39799 | 9950 | 20.0 | 0.0025 |

Table 1: Additional details for binary classification datasets used in convex minimization experiments. Kernel bandwidths were selected by 10-fold cross validation on the training set. SVRG step-sizes were selected by 3-fold CV on the training set. See text for more details.

as a training set and 20 percent split as the test set. The bandwidth parameters for the RBF kernel were selected by grid search using 10-fold cross-validation on the training splits. The grid of kernel bandwidth parameters that were considered is [0.05, 0.1, 0.25, 0.5, 1, 2.5, 5, 10, 15, 20]. For the cross-validation, we used batch L-BFGS to minimize both objectives on the rcv1 and mushrooms datasets, while we used the Coin-Betting algorithm on the larger w8a and ijcnn datasets with mini-batches of 100 examples. In both cases, we ran the optimizers for 100 epochs on each fold. The bandwidth parameters that maximized cross-validated accuracy were selected for our final experiments. The final kernel parameters are given in Table 1, along with additional details for each dataset.

We used the default hyper-parameters for all baseline optimizers used in our other experiments. For PLS, we used the exponential exploration strategy and its default hyper-parameters. Fixed step-size SVRG requires that the step-size parameter to be well-tuned in order to obtain a fair comparison with adaptive methods. To do so, we selected step-sizes by grid search. For each step-size, a 3-fold cross-validation experiment was run on each dataset's training set. On each fold, SVRG was run with mini-batches of size 100 for 50 epochs. Final step-sizes were selected by maximizing convergence rate on the cross-validated loss. The grid of possible step-sizes was expanded whenever the best step-size found was the largest or smallest step-size in the considered grid. We found that the mushrooms, ijcnn, and rcv1 datasets admitted very large step-sizes; in this case, we terminated our grid-search when increasing the step-size further gave only marginal improvement. The final step-sizes selected by this procedure are given in Table 1.

Each optimizer was run with five different random seeds in the final experiment. All optimizers used mini-batches of 100 examples and were run for 35 epochs. Experiment figures display shaded error bars of one standard-deviation from the mean. Note that we did not use a bias parameter in these experiments.

### F.3 Multi-class Classification using Deep Networks

For mutliclass-classification with deep networks, we considered the MNIST and CIFAR10 datasets, each with 10 classes. For MNIST, we used the standard training set consisting of 60k examples and a test set of 10k examples; whereas for CIFAR10, this split was 50k training examples and 10k examples in the test set. As in the kernel experiments, we evaluated the optimizers using the softmax. All optimizers were used with their default learning rates and without any weight decay. We used the experimental setup proposed in [51] and used a batch-size of 128 for all methods and datasets. As before, each optimizer was run with five different random seeds in the final experiment. The optimizers were run until the performance of most methods saturated; 100 epochs for MNIST and 200 epochs for the models on the CIFAR10 dataset. We compare against a tuned SGD method, that uses a constant step-size selected according to a search on the $[1e-1, 1e-5]$ grid and picking the variant that led to the best convergence in the training loss. This procedure resulted in choosing a step-size of 0.01 for the MLP on MNIST and 0.1 for both models on CIFAR10.

## G   Additional Results

| | | |
|---|---|---|
| — SGD(0.001) | — SGD + Armijo(0.2) | — SGD + Armijo(0.01) |
| — Adam(0.01) | — SGD(0.1) | — SGD(0.01) |
| — SGD(0.00001) | — Adam(0.00001) | — Adam(0.001) |
| — Adam(0.1) | — SGD + Armijo(0.5) | — SGD + Armijo(0.1) |

Figure 6: Testing the robustness of Adam, SGD and SGD with Armijo line-search for training ResNet on CIFAR10. SGD is highly sensitive to it's fixed step-size; selecting too small a step-size results in very slow convergence. In contrast, SGD + Armijo has similar performance with $c = 0.1$ and $c = 0.01$ and all $c$ values obtain reasonable performance. We note that Adam is similarly robust to its initial learning-rate parameter.

### G.1   Evaluating robustness and computation

In this experiment, we compare the robustness and computational complexity between the three best performing methods across datasets: Adam, constant step-size and SGD with Armijo line-search. For both Adam and constant step-size SGD, we vary the step-size in the $[10^{-1}, 10^{-5}]$ range; whereas for the SGD with line-search, we vary the parameter $c$ in the $[0.1, 0.5]$ range and vary $\eta_{max} \in [1, 10^3]$ range. We observe that although the performance of constant step-size SGD is sensitive to its step-size; SGD with

Figure 7: Min-max optimization on synthetic bilinear example (left) with interpolation (right) without interpolation. SEG with Lipschitz line-search converges linearly when interpolation is satisfied – in agreement with in Theorem 7 – although it fails to converge when interpolation is violated.

Armijo line-search is robust with respect to the $c$ parameter. Similarly, we find that Adam is quite robust with respect to its initial learning rate.

### G.2 Min-max optimization for bilinear games

Chavdarova et al. [15] propose a challenging stochastic bilinear game as follows:

$$\min_{\boldsymbol{\theta}\in\mathbb{R}^d}\max_{\boldsymbol{\varphi}\in\mathbb{R}^d}\frac{1}{n}\sum_{i=1}^{n}\left(\boldsymbol{\theta}^\top\boldsymbol{b}_i+\boldsymbol{\theta}^\top\boldsymbol{A}_i\boldsymbol{\varphi}+\boldsymbol{c}_i^\top\boldsymbol{\varphi}\right),\ \ [\boldsymbol{A}_i]_{kl}=\delta_{kli}\,,\ [\boldsymbol{b}_i]_k\,,[\boldsymbol{c}_i]_k\sim\mathcal{N}(0,\tfrac{1}{d}),1\le k,l\le d$$

Standard methods such as stochastic extragradient fail to converge on this example. We compare Adam, ExtraAdam [24], SEG with backtracking line-search using Equation 4 with $c = 1/\sqrt{2}$ and $p$-SVRE [15]. The latter combines restart, extrapolation and variance reduction for finite sum. It exhibits linear convergence rate but requires the tuning of the restart parameter $p$ and do not have any convergence guarantees on such bilinear problem. ExtraAdam [24] combines extrapolation and Adam has good performances on GANs although it fails to converge on this simple stochastic bilinear example.

In our synthetic experiment, we consider two variants of this bilinear game; one where interpolation condition is satisfied, and the other when it is not. As predicted by the theory, SEG + Lipschitz results in linear convergence where interpolation is satisfied and does not converge to the solution when it is not. When interpolation is satisfied, empirical convergence rate is faster than SVRE, the best variance reduced method. Note that SVRE does well even in the absence of interpolation, and the both variants of Adam fail to converge on either example.

### G.3 Synthetic Experiment and Binary Classification with Kernels

We provide additional results for binary classification with RBF kernels on the rcv1 and w8a datasets. As before, we do not use regularization. We compare against L4 Mom [68] as well as the original baselines introduced in Section 7.1. For fairness, we reproduce the results for mushrooms and ijcnn with L4 Mom included. Figure 8 shows the training loss and test accuracy for the methods considered, while Figure 10 shows the evolution of step-sizes for SGD+Armijo on all four kernel datasets.

The proposed line-search methods perform well on both rcv1 and w8a although neither dataset satisfies the interpolation condition with the selected kernel bandwidths. Furthermore, all of the proposed line-search methods converge quickly and remain at the global minimum for the w8a dataset, which is particularly ill-conditioned. In contrast, adaptive optimizers, such as Adam, fail to converge. Unlike other methods, PLS uses a separate mini-batch for each step of the line-search procedure. Accordingly, we plot the number of iterations *accepted* by the probabilistic Wolfe conditions, which may correspond to several mini-batches of information. Despite this, PLS converges slowly. In practice, we observed that the initial step-size was accepted at most iterations of the PLS line-search.

Figure 10 provides an additional comparison against L4 Mom on the synthetic matrix factorization problem from Section 7.2. We observe that L4 Mom is unstable when used for *stochastic* optimization of this problem, especially when interpolation is not satisfied. The method converges slowly when interpolation is satisfied.

Figure 8: Binary classification using a softmax loss and RBF kernels on the mushrooms, ijcnn, rcv1, and w8a datasets. *Only* the mushrooms dataset satisfies interpolation with the selected kernel bandwidths. We compare against L4 Mom in addition to the other baseline methods; L4 Mom converges quickly on all datasets, but is unstable on ijcnn. Note that w8a dataset is particularly challenging for Adam, which shows large, periodic drops test accuracy. Our line-search methods quickly and stably converge to the global minimum despite the ill-conditioning.

Figure 9: Variation in the step-sizes for SGD + Armijo for binary classification with softmax loss and RBF kernels on the mushrooms, ijcnn, rcv1 and w8a datasets. Recall that we use reset option 2 in Algorithm 2. The step-size grows exponentially on mushrooms, which satisfies interpolation. In contrast, the step-sizes for rcv1, ijcnn, and w8a increase or decrease to match the smoothness of the problem.

Figure 10: Matrix factorization using the true model and rank 1, 4, 10 factorizations. Rank 1 factorization is under-parametrized, while ranks 4 and 10 are over-parametrized. Only rank 10 factorization and the true model satisfy interpolation. We include L4 Mom as additional baseline optimizer. L4 Mom is unstable and does not converge on rank 1 and 4 factorization, where interpolation is not satisfied; it exhibits slow convergence on the true model and rank 10 factorization.

# H Algorithm Pseudo-Code

---

**Algorithm 3** SGD+Goldstein($f, w_0, \eta_{\max}, b, c, \beta, \gamma$)

---

1: $\eta \leftarrow \eta_{\max}$
2: **for** $k = 0, \ldots, T$ **do**
3:     $i_k \leftarrow$ sample a minibatch of size $b$ with replacement
4:     **while** 1 **do**
5:         **if** $f_{ik}(w_k - \eta \nabla f_{ik}(w_k)) > f_{ik}(w_k) - c \cdot \eta \|\nabla f_{ik}(w_k)\|^2$ **then**       ▷ check Equation (1)
6:             $\eta \leftarrow \beta \cdot \eta$
7:         **else if** $f_{ik}(w_k - \eta \nabla f_{ik}(w_k)) < f_{ik}(w_k) - (1-c) \cdot \eta \|\nabla f_{ik}(w_k)\|^2$ **then**   ▷ check curvature condition
8:             $\eta \leftarrow \min\{\gamma \cdot \eta, \eta_{\max}\}$
9:         **else**
10:            break                            ▷ accept step-size $\eta$
11:         **end if**
12:     **end while**
13:     $w_{k+1} \leftarrow w_k - \eta \nabla f_{ik}(w_k)$                 ▷ take SGD step with $\eta$
14: **end for**
15:
16: **return** $w_{k+1}$

---

---

**Algorithm 4** SEG+Lipschitz($f, w_0, \eta_{\max}, b, c, \beta, \gamma, \texttt{opt}$)

---

1: $\eta \leftarrow \eta_{\max}$
2: **for** $k = 0, \ldots, T$ **do**
3:     $i_k \leftarrow$ sample a minibatch of size $b$ with replacement
4:     $\eta \leftarrow \texttt{reset}(\eta, \eta_{\max}, \gamma, b, k, \texttt{opt})$
5:     **while** $\|\nabla f_{ik}(w_k - \eta \nabla f_{ik}(w_k)) - \nabla f_{ik}(w_k)\| > c \, \|\nabla f_{ik}(w_k)\|$ **do**     ▷ check Equation (4)
6:         $\eta \leftarrow \beta \cdot \eta$                          ▷ backtrack by a multiple of $\beta$
7:     **end while**
8:     $w'_k \leftarrow w_k - \eta \nabla f_{ik}(w_k)$                 ▷ take SEG step with $\eta$
9:     $w_{k+1} \leftarrow w_k - \eta \nabla f_{ik}(w'_k)$
10: **end for**
11:
12: **return** $w_{k+1}$

---

Figure 11: Pseudo-code for two back-tracking line-searches used in our experiments. SGD+Goldstein implements SGD with the Goldstein line search described in Section 6.1 and SEG+Lipschitz implements SEG with the Lipschitz line-search described in Section 5. For both line-searches, we use a simple back-tracking approach that multiplies the step-size by $\beta < 1$ when the line-search is not satisfied. We implement the forward search for Goldstein line-search in similar manner and multiply the step-size by $\gamma > 1$. See Algorithm 2 for the implementation of the reset procedure.

---
**Algorithm 5** `Polyak+Armijo`$(f, w_0, \eta_{\max}, b, c, \beta, \gamma, \alpha, \mathtt{opt})$
---
1:  $\eta \leftarrow \eta_{\max}$
2:  **for** $k = 0, \dots, T$ **do**
3:      $i_k \leftarrow$ sample a minibatch of size $b$ with replacement
4:      $\eta \leftarrow \mathtt{reset}(\eta, \eta_{\max}, \gamma, b, k, \mathtt{opt})$
5:      **while** $f_{ik}(w_k - \eta\nabla f_{ik}(w_k)) > f_{ik}(w_k) - c \cdot \eta \left\| \nabla f_{ik}(w_k) \right\|^2$ **do**          $\triangleright$ check Equation (1)
6:          $\eta \leftarrow \beta \cdot \eta$          $\triangleright$ backtrack by a multiple of $\beta$
7:      **end while**
8:      $w_{k+1} \leftarrow w_k - \eta\nabla f_{ik}(w_k) + \alpha(w_k - w_{k-1})$          $\triangleright$ take SGD step with $\eta$ and Polyak momentum
9:  **end for**
10:
11: **return** $w_{k+1}$
---

---
**Algorithm 6** `Nesterov+Armijo`$(f, w_0, \eta_{\max}, b, c, \beta, \gamma, \mathtt{opt})$
---
1:  $\tau \leftarrow 1$          $\triangleright$ bookkeeping for Nesterov acceleration
2:  $\lambda \leftarrow 1$
3:  $\lambda_{\mathrm{prev}} \leftarrow 0$
4:
5:  $\eta \leftarrow \eta_{\max}$
6:  **for** $k = 0, \dots, T$ **do**
7:      $i_k \leftarrow$ sample a minibatch of size $b$ with replacement
8:      $\eta \leftarrow \mathtt{reset}(\eta, \eta_{\max}, \gamma, b, k, \mathtt{opt})$
9:      **while** $f_{ik}(w_k - \eta\nabla f_{ik}(w_k)) > f_{ik}(w_k) - c \cdot \eta \left\| \nabla f_{ik}(w_k) \right\|^2$ **do**          $\triangleright$ check Equation (1)
10:          $\eta \leftarrow \beta \cdot \eta$          $\triangleright$ backtrack by a multiple of $\beta$
11:      **end while**
12:      $w'_k \leftarrow w_k - \eta\nabla f_{ik}(w_k)$
13:      $w_{k+1} \leftarrow (1 - \tau) \cdot w'_k + \tau \cdot w_k$          $\triangleright$ Nesterov accelerated update with $\eta$
14:
15:      $\mathtt{temp} \leftarrow \lambda$          $\triangleright$ bookkeeping for Nesterov acceleration
16:      $\lambda \leftarrow \left( 1 + \sqrt{1 + 4\lambda_{\mathrm{prev}}^2} \right) / 2$
17:      $\lambda_{\mathrm{prev}} \leftarrow \mathtt{temp}$
18:      $\tau \leftarrow (1 - \lambda_{\mathrm{prev}}) / \lambda$
19: **end for**
20:
21: **return** $w_{k+1}$
---

Figure 12: Pseudo-code for using Polyak momentum and Nesterov acceleration with our proposed line-search techniques. `Polyak+Armijo` implements SGD with Polyak momentum and Armijo line-search and `Nesterov+Armijo` implements SGD with Nesterov acceleration and Armijo line-search. Both methods are described in 6.2. See Algorithm 2 for the implementation of the `reset` procedure.