[Reviews · NeurIPS 2019]

Reviewer 1



1) see above 2) well written 3) nice (though not overwhelming) simulations 4) the names of the authors were on the supplementary material...

Reviewer 2



UPDATE: I've read the other reviews and the rebuttal. I am keeping my score - this is a good paper. --------------------------------------------------- Originality. The study of Stochastic Gradient Descent in overparametrized setting is a popular and important trend in a recent development of huge-scale optimization for deep-learning. The authors propose a very basic and classical method, consisting from the well-known algorithmical blocks (SGD + Armijo-type line search) together with its first theoretical justification under "interpolation assumption". The proof of convergence (for example, Theorem 2) mainly consists from the standard arguments (which are used for the proof of the classical non-stochastic Gradient Method under Lipschitz-continuous gradients). Usually, if we try to repeat this analysis with SGD, the main struggle is how to take the expectation. Under the interpolation assumption, the authors are able to take this expectation easily and therefore propagate non-stochastic GD rates to the stochastic case. Despite the fact, that presented analysis is simple, up to my knowledge, these theoretical results and corresponding assumption are very novel. The main concern about this would be to make this trick more transparent, probably, to provide some insight about the proof in the main paper. It would make the presentation more attractive, if to declare additionally the limitations of this proof, with a discussion of possible relaxation of the assumption. From now it is also not very clear, how restrictive is it. In my opinion, more examples when interpolation assumption is satisfied, stated precisely and in the paper would help. Quality. The submission contains a number of proven theorems with results about convergence of the proposed method. The paper is self-contained. Numerical experiments are supported with the code. One more concern would be a lack of comparison of the results with that one, given in works [5] and [77] (which are only briefly mentioned in the paper). Clarity. The paper is written in a well-mannered way, and it is easy to follow. It informs the reader adequately about the field and gives a comprehensive reference. It would be interesting to see more examples and possible generalization of the presented approach. For clarity of the text, I would also prefer to have the assumption ("interpolation" and "SGC") stated in a kind of separated form, not between the main text. Significance. Provided fast convergence rates appears to be the first strong theoretical result about adaptive SGD (to the best of my knowledge), making the results of this work significant.

Reviewer 3



EDIT: The author feedback contained rigorous corrections to proofs that were incorrect, as well as experiments showing that their method does not lead to more function evaluations. These were exactly my two biggest issues, and so I have changed my score from a 4 to a 7. The previous score of 4 was not due to the significance, impact or novelty, but just because I highly value rigor. With the corrections, I believe this is a very solid paper. -------------------------- The paper's strongest contribution is its novelty. The authors do a good job of taing a classical method (the Armijo line search), and deriving a version that is not just applicable, but useful in modern day machine learning. Moreover, the authors do a good job of motivating the use of this method in contrast to popular methods with difficult hyperparameter tuning, such as Adam. The authors do a great job in citing relevant work, and telling a comprehensive story about the Armijo line search, up to their version today, as well as tying it in to modern research themes such as overparameterization and variance reduction. The quality of the paper unfortunately varies between different parts. The paper is generally well-written, well-motivated, and very clear in its presentation of its theoretical and experimental contributions. Unfortunately, some of the proofs are either wrong or are not rigorous. Theorem 1 contains multiple steps that are flawed. In particular, at one point the authors seem to say that the expected value of a minimum is equal to the minimum of expected values. Even simple settings of the \mu_i, L_i, and \eta_max can be used to disprove some of the equations in this proof where there are interactions between mins, maxes, and expected values. Even assuming these issues are fixed, the authors then seem to use (without proof) the notion that if f_1, ..., f_n are each \mu_i-strongly convex and L_i-smooth, and the average f of these functions is \mu-strongly convex and L-smooth, then the average of \mu_i/L_i is less than or equal to \mu/L. It is not clear to me that this is true, and even if it were, there would need to be a proof of this fact. As it stands, the proof of Theorem 1 is incorrect, and may even need the statement of the theorem to be reworked (to be more similar to Theorem 4) in order to hold. Issues with taking the expected value of a minimum also appear in E.2.2, though it seems the authors take some steps there to better ensure that the equations hold. However, this section could use more clarity on this, due to the similar issues above. Theorem 3 is more problematic in general. In the proof, the authors take 1st order Taylor expansions, and then simply remove the 2nd order error terms by stating that the step-size is sufficiently small. While they acknowledge that they do this, this does not change the fact that this still isn't a rigorous proof. The authors never characterize how small the step-size must be in order to discount such terms. Moreover, if the step-size is too small, then it will be impossible to derive fast convergence rates. Thus, as stated, Theorem 3 is simply not proven. While I believe it could be proven by using the intuitive idea of making the 2nd order error terms sufficiently small, the authors do not do this. They regularly state that a function equals its 1st order Taylor expansion, which just isn't true. Given that the authors list Theorem 3 as one of their main contributions, this greatly detracts from the quality of the paper. In general, the proofs in the appendix could also be better written. In addition to omitting certain details, the overall format of the proofs is a little inconsistent, and could use some more careful explanation in many parts (eg. using more equations, instead of simply saying things such as "subtracting the above terms" as in the proof of Theorem 3). For example, the writing and explanation in the proof of Theorem 7 is much better and clearer than the writing and explanation in the proofs of Theorems 1-3. The experiments are solid, and encompass a variety of different machine learning tasks, which is good. The authors also do a good job of stating how each method compared is initialized, as well as how they are evaluated. That being said, there are issues that could be improved upon. First, given the very noisy nature of the plots in Figure 3, it would be hugely beneficial to include error bar plots, especially as the authors plot the average of 5 independent runs. Second, the line search method requires more function evaluations, something that the authors discuss at length. However, this may increase the amount of compute time required per iteration. While the authors state that they verify that the number of function evaluations is at most twice the number of gradient evaluations, it would be good to see this represented in a plot. For example, one simple thing would be to plot the error of various training methods with respect to compute time. This would give the reader a clearer picture of the actual runtime considerations involved in the authors' method. The main body of the paper is very clear. The authors do a good job of organizing their work according to different theoretical and practical concerns, and do a good job of educating their readers throughout. Moreover, the authors deserve commendation for being clear about how the methods are analyzed in theory versus how they are implemented in practice. The experiments are all well-detailed and clear, as is the supplied code. The work has mixed significance. Based on the abstract alone, the work is very significant, both to theoreticians and practitioners. The work advances the idea of interpolation as a useful condition for various optimization problems, and devises and easily implementable, widely useful method for training machine learning models. However, the significance of the work's theory is marred by the incorrect and non-rigorous proofs. Moreover, the practical significance could be improved by further demonstrating that the line search methods do not increase the run time needed to reach a certain accuracy.

[Author Response · NeurIPS 2019]

We thank the reviewers for their time and helpful suggestions, which we will use to improve the paper's presentation.

**R1, R2 (relevance of interpolation):** Please refer to lines 39-42 for examples where interpolation is satisfied. Recent
work [5,7,42,46,77] views interpolation as key to understanding the effectiveness of SGD for deep learning. Moreover,
we utilized this assumption to make algorithmic contributions that result in better empirical performance.

**R2 (comparison with [5], [77]):** They both use a constant step-size of $\frac{1}{L}$, which is either unknown or gives an
overly-conservative, small step-size. Our initial experiments confirmed that it lead to worse empirical performance and
we will mention this.

**R2, R3 (wall-clock):** For the line-search, we did ensure that the number of additional function evaluations is not large
(Section 7). In the Fig. 1 below, we show the wall-clock time per iteration averaged across training for the three datasets.

**R3 (error bars):** The figures in Section 7.3 do have error bars, but they unfortunately look like spikes in the submitted
version. We include one figure below with clearer error bars and will similarly update the remaining figures.

**R3 (Fixing typo for Theorem 1):** We correct the proof and statement of Theorem 1 below. Starting from the line
justified by Equation 2 in Appendix B (recall that $\mu_{ik} = 0$ if the $f_{ik}$ is not strongly-convex),

$$\mathbb{E}\left[\|w_{k+1} - w^*\|^2\right] \leq \left(1 - \mathbb{E}_{ik}\left[\mu_{ik} \ \min\left\{\frac{1}{L_{ik}}, \eta_{\max}\right\}\right]\right)\|w_k - w^*\|^2$$

We consider the following two cases: $\eta_{\max} < 1/L_{\max}$ and $\eta_{\max} \geq 1/L_{\max}$. When $\eta_{\max} < 1/L_{\max}$,

$$\mathbb{E}\left[\|w_{k+1} - w^*\|^2\right] \leq (1 - \mathbb{E}_{ik}\left[\mu_{ik} \ \eta_{\max}\right])\|w_k - w^*\|^2 = (1 - \bar{\mu} \ \eta_{\max})\|w_k - w^*\|^2$$

When $\eta_{\max} \geq 1/L_{\max}$, we use $\min\left\{\frac{1}{L_{ik}}, \eta_{\max}\right\} \geq \min\left\{\frac{1}{L_{\max}}, \eta_{\max}\right\}$ to obtain

$$\mathbb{E}\left[\|w_{k+1} - w^*\|^2\right] \leq \left(1 - \mathbb{E}_{ik}\left[\mu_{ik} \ \frac{1}{L_{\max}}\right]\right)\|w_k - w^*\|^2 = \left(1 - \frac{\bar{\mu}}{L_{\max}}\right)\|w_k - w^*\|^2 .$$

Combining the two cases gives us the theorem statement with $L_{max}$ instead of $L$. We will make this change in
Theorem 1 statement. Note that Theorem 4's proof will be changed similarly.

**R3 (Requested) rigorous proof for Theorem 3:** We can prove an $O(1/T)$ rate by bounding $\eta_{\max} \leq \frac{3}{2\rho L}$ as follows:

$$\frac{f(w_{k+1}) - f(w_k)}{\eta_k} \leq \frac{L\eta_k}{2}\|\nabla f_{ik}(w_k)\|^2 - \langle\nabla f(w_k), \nabla f_{ik}(w_k)\rangle \qquad \text{(Using smoothness and dividing by } \eta_k)$$

$$\implies \mathbb{E}\left[\frac{f(w_{k+1}) - f(w_k)}{\eta_k}\right] \leq \left(\frac{L\eta_{\max}\rho}{2} - 1\right)\|\nabla f(w_k)\|^2 \qquad \text{(Since } \eta_k \leq \eta_{\max} \text{ and using the SGC)}$$

$$\|\nabla f(w_k)\|^2 \leq \frac{1}{1 - \frac{L\eta_{\max}\rho}{2}}\mathbb{E}\left[\frac{f(w_k) - f(w_{k+1})}{\eta_k}\right] \qquad \text{(Rearranging and upper-bounding } \eta_{\max} \leq \frac{2}{L\rho}\text{,)}$$

$$\implies \|\nabla f(w_k)\|^2 \leq \left(\frac{1}{1 - \frac{L\eta_{\max}\rho}{2}}\right)\left(\frac{1}{\eta_{\max}} + \frac{L_{max}}{2(1-c)}\right)\mathbb{E}\left[f(w_k) - f(w_{k+1})\right] .$$

$$\text{(Bounding } \eta_k \text{ using the line-search similar to Appendix C)}$$

Telescoping and setting $c = 1/2$ and $\eta_{\max} \leq \frac{3}{2\rho L}$ completes the proof. It is non-trivial to avoid the dependence of $\rho, L$ in
$\eta_{\max}$ and we leave it as future work. Regardless of this result, we believe that this paper's contributions are impactful.

Figure 1: **Left**: CIFAR-10 with new error-bar style. **Right**: Average iteration times on CIFAR-10.



[Meta-Review · NeurIPS 2019]

This paper brings a classic idea into the present and makes progress on a vexing problem with SGD --- setting the step size. The authors provide theoretical evidence as well as emipirical evidence that their method is useful. The assumptions may be somewhat limiting; one version requires strong convexity and when that is relaxed, other assumptions must be made. But this work points to a path that may be useful in the long-run. An important way of contribution in ML is bridging fields; that could mean bringing in ideas that are state-of-the-art in other fields or it could mean revisiting classic ideas in new ways. Indeed, SGD itself is a revisitation of a classic idea that was impractical in its own time, but found wide applications when data sets grew large. This paper is a good contribution because it bridges fields and provides rigorous evidence to support their improvements. I'm strongly in favor of acceptance.